# Regret-Guided Search Control for Efficient Learning in AlphaZero

**Yun-Jui Tsai[1, 2], Wei-Yu Chen[2, 3], Yan-Ru Ju[2], Yu-Hung Chang[2], Ti-Rong Wu[2†]**

[1]Department of Computer Science, National Yang Ming Chiao Tung University, Taiwan
[2]Institute of Information Science, Academia Sinica, Taiwan
[3]College of Computing, Georgia Institute of Technology

```
b08202011.cs12@nycu.edu.tw, wchen784@gatech.edu
{yanru, yuhung, tirongwu}@iis.sinica.edu.tw
```

## Abstract

Reinforcement learning (RL) agents achieve remarkable performance but remain far less learning-efficient than humans. While RL agents require extensive self-play games to extract useful signals, humans often need only a few games, improving rapidly by repeatedly revisiting states where mistakes occurred. This idea, known as *search control*, aims to restart from valuable states rather than always from the initial state. In AlphaZero, prior work Go-Exploit applies this idea by sampling past states from self-play or search trees, but it treats all states equally, regardless of their learning potential. We propose *Regret-Guided Search Control* (RGSC), which extends AlphaZero with a regret network that learns to identify high-regret states, where the agent's evaluation diverges most from the actual outcome. These states are collected from both self-play trajectories and MCTS nodes, stored in a prioritized regret buffer, and reused as new starting positions. Across 9x9 Go, 10x10 Othello, and 11x11 Hex, RGSC outperforms AlphaZero and Go-Exploit by an average of 77 and 89 Elo, respectively. When training on a well-trained 9x9 Go model, RGSC further improves the win rate against KataGo from 69.3% to 78.2%, while both baselines show no improvement. These results demonstrate that RGSC provides an effective mechanism for search control, improving both efficiency and robustness of AlphaZero training. Our code is available at https://rlg.iis.sinica.edu.tw/papers/rgsc.

## 1 Introduction

Reinforcement learning (RL) is the process of training an agent through interaction with the environment and optimizing its behavior based on rewards. The foundations of RL were originally inspired by human learning, where humans acquire new knowledge through trial-and-error experiences. However, despite this conceptual similarity, current RL approaches remain far less efficient than human learning (Tsividis et al., 2021; Iii & Sadigh, 2023). Consider the case of mastering the game of Go. An RL agent such as AlphaZero (Silver et al., 2017; 2018) requires millions of self-play games to reach superhuman performance. In contrast, professional human players can achieve comparable strength after far fewer games.

One key difference lies in how learning progresses. As illustrated in Figure 1, humans do not rely on playing massive numbers of games from the beginning. Instead, they repeatedly review the critical positions where mistakes occurred and refine their understanding until those weaknesses are corrected. AlphaZero, in contrast, always restarts from the empty board and updates all positions uniformly based on the obtained outcome, which substantially increases the number of episodes required to master a game.

---

[†]Corresponding author.

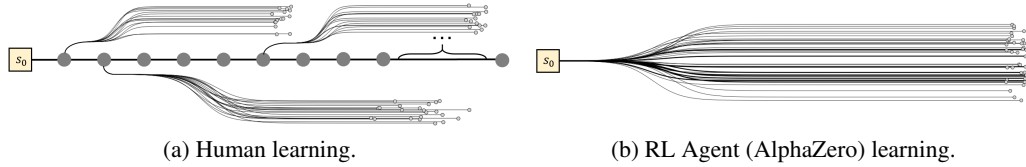

(a) Human learning.                    (b) RL Agent (AlphaZero) learning.

Figure 1: Humans focus on correcting mistakes, whereas RL always starts from the initial state.

To bridge this gap, recent studies have investigated restarting strategies to improve the efficiency of RL. This idea, originating from Sutton & Barto (2018), was formalized as the concept of *search control*, which refers to selecting critical starting states for the simulated experiences. Building on this principle, several works have been proposed for constructing restart distributions in RL, such as sampling from the past trajectories (Tavakoli et al., 2020), starting from states closer to the goal (Florensa et al., 2017), or leveraging expert demonstrations (Uchendu et al., 2023). Go-Exploit (Trudeau & Bowling, 2023) further extends this idea to the AlphaZero framework by maintaining a buffer of states from self-play trajectories or search nodes and uniformly sampling them as new starting positions. Collectively, these approaches demonstrate that choosing starting states, rather than always restarting from the initial state, can significantly accelerate RL learning. However, a key limitation of Go-Exploit is that it considers all states equally. In practice, not all states contribute equally to learning progress. Many states are already mastered, while only a small subset of states are actually critical for improvement. This phenomenon becomes exacerbated in the later stages of training, as the agent's understanding of the game improves and mistakes become increasingly rare. This motivates the need to identify and prioritize the most informative states for search control.

To address this challenge, we propose *Regret-Guided Search Control* (RGSC), a framework that extends AlphaZero by identifying and revisiting high-regret states. Specifically, RGSC leverages a regret network to detect states where the agent's evaluation diverges most from the game outcome. Since most states have near-zero regret, making direct learning of regret values challenging, we design a ranking-based objective that guides the network to distinguish the most informative states. These states are then stored in a prioritized regret buffer. By repeatedly restarting from these states, the agent can focus on correcting its most critical mistakes, thereby mimicking human learning and achieving more efficient training. Experimental results show that RGSC outperforms both AlphaZero and Go-Exploit across three board games, including 9x9 Go, 10x10 Othello, and 11x11 Hex, achieving an average improvement of 77 Elo over AlphaZero and 89 Elo over Go-Exploit. Furthermore, when continuing training from a strong, nearly converged model in 9x9 Go for 40 iterations, RGSC still improves the win rate from 69.3% to 78.2%, whereas both AlphaZero and Go-Exploit show no improvement. Moreover, additional analysis demonstrates that RGSC successfully identifies high-regret states and systematically reduces their regret during training. In summary, RGSC provides an effective mechanism for search control in AlphaZero. Our results highlight regret-guided search control as a promising direction for improving the efficiency and robustness of reinforcement learning.

## 2 BACKGROUND

### 2.1 SEARCH CONTROL IN REINFORCEMENT LEARNING

The concept of *search control* (Sutton & Barto, 2018) was first introduced in the Dyna general framework (Sutton, 1991), which integrates real experience with model-generated simulated experience. In this setting, simulated experience is generated through search control, which determines the starting states and actions for rollouts, rather than always beginning from a fixed initial state. This allows planning to focus computation on states that provide more information to accelerate learning.

Several subsequent works have adopted the principle of search control by choosing different starting states during training. For example, Go-Explore (Ecoffet et al., 2021) addresses hard-exploration problems by maintaining a database of promising states, and periodically selecting from these states to discover high-reward trajectories. This approach allows systematic exploration of rarely visited regions and achieved state-of-the-art results in an extremely difficult environment, *Montezuma's Re-*

*venge.* (Florensa et al., 2017) propose another approach by selecting starting states near the goal and gradually moving them backward, thereby constructing a curriculum in reverse to facilitate learning in sparse reward environments. Jump-Start Reinforcement Learning (JSRL) (Uchendu et al., 2023) samples initial states from expert demonstration trajectories, allowing the agent to focus on meaningful states early in training, thereby improving sample efficiency. Tavakoli et al. (2020) provides a formal definition for exploring restart distributions by introducing a restart distribution $\rho(s)$ over states. By altering the distribution of the restart states, the learning objective is modified to

$$L(\mathbf{w}) \doteq \sum_{s \in \mathcal{S}} \rho(s) \sum_{a \in \mathcal{A}} \pi(a \mid s) \left( q_\pi(s, a) - \hat{q}_\pi(s, a) \right)^2, \tag{1}$$

where $\mathcal{S}$ and $\mathcal{A}$ denote the state and action spaces, $\pi(a \mid s)$ is the policy, $q_\pi(s, a)$ is the true action-value function under $\pi$, and $\hat{q}_\pi(s, a)$ is its learned approximation. The restart distribution $\rho(s)$ specifies the probability of selecting state $s \in \mathcal{S}$ as a restart point. Two restart strategies are proposed: (a) uniform restart, which samples from recent experiences, and (b) prioritized restart, which ranks states according to their state-value temporal-difference (TD) error.

Moreover, search control is also widely applied at the level of task selection under curriculum-based environments. For example, several studies (Jiang et al., 2021; Dennis et al., 2020; Parker-Holder et al., 2022) adaptively sample training levels based on estimated regret, encouraging the agent to focus on levels with higher learning potential. While these methods operate at the level of task selection, our goal is to identify the most challenging states within the same game level.

## 2.2 SEARCH CONTROL IN ALPHAZERO

AlphaZero (Silver et al., 2018) is a reinforcement learning algorithm that can master board games such as Chess, Shogi, and Go without requiring human knowledge. The training process alternates between two phases: a *self-play* phase and an *optimization* phase. During the self-play phase, the agent generates games against itself by combining Monte Carlo tree search (MCTS) Browne et al. (2012); Coulom (2007) with a two-head neural network, including a policy network that outputs a probability distribution over all possible actions and a value head that predicts the win rate of a given state. In the optimization phase, trajectories collected from self-play are stored in a replay buffer and sampled to update the neural network, training the policy head to predict the MCTS search distribution and the value head to predict the final game outcome. Although AlphaZero has demonstrated superhuman performance in board games, it requires extensive computation, especially in games with long trajectories, because every self-play game must start from the empty board. This issue is exacerbated in 19x19 Go, where a single game often exceeds 250 moves, making it necessary to spend enormous computational resources to generate self-play games (e.g., roughly 1.5 million TPU-hours as reported in (Silver et al., 2018)).

To alleviate this issue, several studies have incorporated search control into AlphaZero by adjusting self-play games to begin from particular intermediate states. For example, KataGo (Wu, 2020), one of the current strongest open-source Go programs, proposes selecting starting states either by randomly playing several moves with the policy network or by sampling and slightly modifying states from past self-play trajectories. Similar to Florensa et al. (2017), Björnsson (2023) proposes starting self-play from later stages of the game and gradually shifting the starting state toward the initial position. This approach accelerates the training process, particularly in the early phases. Recently, Trudeau & Bowling (2023) proposes Go-Exploit, which systematically investigates restart state methods within the AlphaZero algorithm. Go-Exploit maintains a buffer of states collected either from self-play trajectories (Go-Exploit Visited states Circular archive; GEVC) or from nodes within the MCTS (Go-Exploit Search states Circular archive; GESC). For each self-play game, the agent starts from the initial state with probability $\lambda$; otherwise, it uniformly samples a state from the buffer as the starting state. Go-Exploit achieves higher sample efficiency and stronger performance than AlphaZero in both Connect Four and 9x9 Go, with GEVC and GESC showing similar results. However, a key limitation of Go-Exploit is its uniform sampling. By treating all states equally, the method fails to align with the principle of restart distribution mentioned in Equation 1, which emphasizes prioritizing important states that provide better learning.

## 3 REGRET-GUIDED SEARCH CONTROL

### 3.1 REGRET DEFINITION IN BOARD GAMES

We propose *Regret-Guided Search Control* (RGSC), a framework that extends AlphaZero by identifying and prioritizing high-regret states as search control openings for self-play in board games, as shown in Figure 2. Unlike the original AlphaZero, as shown in Figure 1b, where self-play always starts from the empty board, RGSC guides self-play to begin from states with higher *regret*, where regret reflects positions that the current agent has not yet mastered. These states can appear either along the self-play trajectory or within the MCTS search tree. This allows the agent to focus on learning and exploring unfamiliar states with greater potential for improvement. Note that Go-Exploit adopts a similar idea of restarting from previously collected states, but it samples them uniformly, which fails to capture the most informative states.

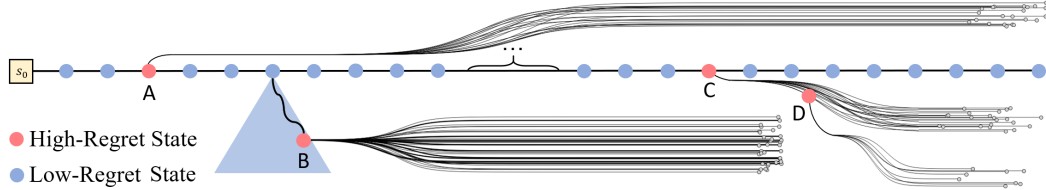

Figure 2: Overview of RGSC. The regret network selects high-regret states (red circles) from both self-play (`A`) or MTCS search node (`B`), which serve as restart points for further self-play. The newly generated trajectories can further branch out, e.g., state `D` originates from a restart at state `C`.

To formalize this idea, we define regret as a measure of the discrepancy between the agent's evaluation and the game outcome. Given a self-play trajectory with state $s_0$, $s_1$, ..., $s_T$ and a game outcome $z$, the regret of state $s_t$ is

$$\mathcal{R}(s_t) = \frac{1}{T-t} \sum_{i=t}^{T} \left( V_{selected}(s_i) - z \right)^2, \tag{2}$$

where $V_{selected}(s_i)$ represents the MCTS value of the selected action at state $s_i$. Intuitively, $\mathcal{R}(s_t)$ measures the average discrepancy accumulated from $s_t$ to the terminal state $s_T$, capturing states whose mis-evaluation has long-term impact on the outcome.

Note that $\mathcal{R}(s_t)$ is calculated only after the game is finished. Moreover, the same state $s$ may have different regret values across trajectories, since the subsequent moves and outcome can vary. As training progresses and the agent's evaluations become more accurate, the regret of previously misjudged states gradually decreases. Conceptually, this resembles how human players repeatedly review their mistakes to improve.

### 3.2 REGRET NETWORK

Although the regret $\mathcal{R}(s_t)$ of states on a finished self-play trajectory can be directly calculated, many internal states in the MCTS search tree are not part of the actual trajectory and thus have no regret values. Nevertheless, these states may still include critical states that the agent has not yet mastered. Leveraging such states allows the agent to obtain more diverse restart states beyond the limited set of self-play trajectories.

A naive approach is to train a *regret value network* that, given a state, directly predicts its regret value, similar to settings in learning-to-learn problems (Wang et al., 2017; Chu et al., 2024; Gupta et al., 2020). However, predicting regret value for arbitrary states in AlphaZero training is highly challenging. First, the distribution is extremely imbalanced: most states have near-zero regret, while high-regret states occur only rarely. Second, the learning target is non-stationary: high-regret states are selected for restarts, and once revisited, their regret typically decreases quickly as the agent corrects its mistakes. As a result, predicting regret becomes extremely difficult for this naive approach.

To tackle this challenge, we propose to learn regret with a ranking-based objective. The key idea is that instead of predicting precise regret values, which are imbalanced and non-stationary targets,

we only need to identify which states have higher regret among all collected states. This relaxation guides the model to focus on the most informative states, ensuring that they are included in the prioritized buffer for restarting self-play.

Specifically, we incorporate a *regret ranking network* into the AlphaZero network. Given a state $s$, the regret ranking network outputs an unnormalized score, $\gamma_s$, where $\gamma_s$ represents the ranking score of state $s$. Note that $\gamma$ is a relative ranking score rather than the true regret value, with higher scores corresponding to states with higher regrets. For a set of candidate states $\mathcal{S}$, the restart distribution $\rho(s \mid \mathcal{S})$ is derived as follows:

$$\rho(s \mid \mathcal{S}) = \frac{\exp(\gamma_s)}{\sum_{s' \in \mathcal{S}} \exp(\gamma_{s'})}. \tag{3}$$

Following Equation 1, the ranking objective is to maximize

$$\mathcal{J}_{\text{rank}} = \sum_{s \in \mathcal{S}} \rho(s \mid \mathcal{S}) \mathcal{R}(s), \tag{4}$$

which encourages the model to assign high probability to the highest-regret states, as these are the most critical for maximizing $\mathcal{J}_{\text{rank}}$ and for restarting self-play.

To better optimize the regret ranking network, we apply an exponential transformation to the regret values, which preserves the ranking order. Then, the network is optimized by using a surrogate objective

$$\tilde{\mathcal{J}}_{\text{rank}} = \sum_{s \in \mathcal{S}} \rho(s \mid \mathcal{S}) \exp\big(\mathcal{R}(s)\big), \tag{5}$$

and the corresponding loss function is defined as

$$\mathcal{L}_{\text{rank}} = -\log \tilde{\mathcal{J}}_{\text{rank}} = -\log \sum_{s \in \mathcal{S}} \rho(s \mid \mathcal{S}) \exp\big(\mathcal{R}(s)\big) \tag{6}$$

$$= -\log \sum_{s \in \mathcal{S}} \Big( \exp\big(\log \text{softmax}(\gamma_s) + \mathcal{R}(s)\big)\Big). \tag{7}$$

The derived loss can be interpreted as adding regret as an additive bias to the log-softmax scores, providing a smooth approximation to selecting the highest-regret states. The following is the step-by-step derivation:

$$\mathcal{L}_{\text{rank}} = -\log \sum_{s \in \mathcal{S}} \rho(s \mid \mathcal{S}) \exp\big(\mathcal{R}(s)\big) \tag{8}$$

$$= -\log \left( \sum_{s \in \mathcal{S}} \frac{\exp(\gamma_s)}{\sum_{s' \in \mathcal{S}} \exp(\gamma_{s'})} \exp\big(\mathcal{R}(s)\big) \right) \tag{9}$$

$$= -\log \sum_{s \in \mathcal{S}} \left( \exp\left( \log\Big(\frac{\exp(\gamma_s)}{\sum_{s' \in \mathcal{S}} \exp(\gamma_{s'})}\Big) \right) \exp\big(\mathcal{R}(s)\big) \right) \tag{10}$$

$$= -\log \sum_{s \in \mathcal{S}} \left( \exp\left( \log\Big(\frac{\exp(\gamma_s)}{\sum_{s' \in \mathcal{S}} \exp(\gamma_{s'})}\Big) + \mathcal{R}(s) \right) \right) \tag{11}$$

$$= -\log \sum_{s \in \mathcal{S}} \left( \exp\left( \log\Big(\text{softmax}(\gamma_s)\Big) + \mathcal{R}(s) \right) \right) \tag{12}$$

Although the regret ranking network can differentiate states with higher regrets, its ranking score is not bounded within the true regret value range. Therefore, to provide a quantitative measurement of regret for the selected states, our regret network consists of both a regret value network and a regret ranking network. The regret ranking network identifies high-regret states, while the regret value network estimates their actual regret value.

### 3.3 PRIORITIZED REGRET BUFFER FOR SEARCH CONTROL

We describe the *prioritized regret buffer* (PRB), which utilizes the regret network to allow search control during AlphaZero training. For each self-play game, we first apply the regret ranking network to evaluate all states that appear both in the self-play trajectory and in the MCTS search trees.

The state with the highest ranking score is then selected. If the selected state $s$ appears in the self-play trajectory, we calculate its regret value $\mathcal{R}(s)$ using Equation 2; if it appears only in the search tree, its regret value is estimated by the regret value network. The PRB maintains only a fixed capacity of $K$ states. If the PRB is not yet full, the selected state $s$ is added directly. Otherwise, it is added only if its regret is higher than that of the lowest-regret state currently in the PRB. This ensures that the PRB consistently stores a set of high-regret states for restarting.

For each self-play game, search control guides the choice of restarting state, starting from the empty board with probability $1 - \lambda$, and from a state sampled from PRB with probability $\lambda$. We adopt a softmax distribution over all states in PRB when sampling to ensure high-regret states are prioritized. The probability of selecting a state $s_i$ in PRB is defined as $P(s_i) = \mathcal{R}(s_i)^{1/\tau} / \sum_j \mathcal{R}(s_j)^{1/\tau}$, where $\tau$ is the sampling temperature.

For restarting games from states in PRB, we update their regret values $\mathcal{R}^{\text{new}}(s_i)$ after replaying each game using an exponential moving average (EMA):

$$\mathcal{R}^{\text{new}}(s_i) \leftarrow (1 - \alpha) \times \mathcal{R}^{\text{old}}(s_i) + \alpha \times \mathcal{R}(s_i), \tag{13}$$

where $\mathcal{R}^{\text{old}}(s_i)$ is the previous regret value stored in the buffer, $\mathcal{R}(s_i)$ is the regret calculated from the newly finished self-play game, and $\alpha$ is the EMA coefficient. This prevents regret values from decreasing abruptly and ensures that once the agent has consistently mastered this state, its regret will gradually decay, thereby reducing the probability of the state being sampled from the buffer. In summary, this design mirrors how humans repeatedly review mistakes until they are fully understood. We have also provided a detailed algorithm for RGSC in the Appendix B.

## 4 EXPERIMENTS

### 4.1 TOY EXAMPLE

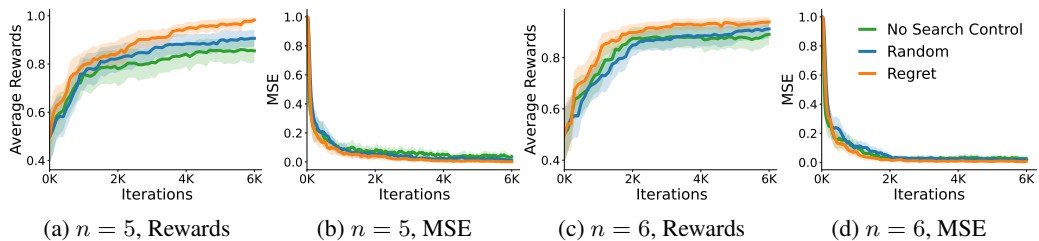

(a) $n = 5$, Rewards    (b) $n = 5$, MSE    (c) $n = 6$, Rewards    (d) $n = 6$, MSE

Figure 3: A toy example on $n$-level binary tree. (a) and (c) show the average rewards during the training, while (b) and (d) show the mean squared error (MSE) of the optimal Q-values during the training. The shaded area is a 95% confidence interval for the mean.

We first investigate search control in a toy environment, an $n$-level sparse-reward binary tree, where each leaf node is assigned an expected reward value $p \in [0, 1]$. The agent starts from the root and selects nodes until reaching a leaf. Upon reaching a leaf node, it receives a stochastic binary reward: 1 with probability $p$ and 0 with probability $1 - p$. Among all leaf nodes, exactly one node is assigned to $p = 1$; thus, the objective in this environment is to discover the unique path that always guarantees a reward 1. Next, we train Q-learning on this environment with three search control methods: (a) No search control, always starting from the root; (b) Random, uniformly sampling from visited nodes; and (c) Regret, sampling nodes in proportion to their regret. For the regret, we simply use $|\hat{Q}(s) - Q(s, a)|$, where $\hat{Q}(s)$ is the empirical maximum expected value estimated from all child nodes, and $Q(s, a)$ is the current Q-value of state $s$ with action $a$. Figure 3 shows the results for the 5- and 6-level binary trees. The Regret method achieves higher average rewards than both Random and No search control. These results demonstrate the importance of prioritizing states with high learning potential and show the effectiveness of the regret-guided search control. Detailed settings of the toy environment are provided in Appendix C.

## 4.2 RGSC IN BOARD GAMES

We compare RGSC against two baseline methods: (a) AlphaZero, which is trained without search control, and (b) Go-Exploit with its GEVC variant described in subsection 2.2, across three board games, including 9x9 Go, 10x10 Othello, and 11x11 Hex. All methods use a 3-block residual network (He et al., 2016) and 200 MCTS simulations per move during self-play. Training runs for 300 iterations, with 160,000 states collected per iteration in 9x9 Go (due to its higher complexity) and 120,000 states in the other two games. We fix the number of training states rather than the number of self-play games per iteration to ensure fairness, since AlphaZero without search control requires more computation to generate a self-play game. Although RGSC requires training two additional heads, the computational overhead is minimal. In particular, as the number of blocks increases, the additional cost becomes negligible. We provide detailed training settings and analyses of the computational cost in subsection A.1 and subsection A.2, respectively. In summary, each training requires approximately 150 NVIDIA RTX A6000 GPU hours.

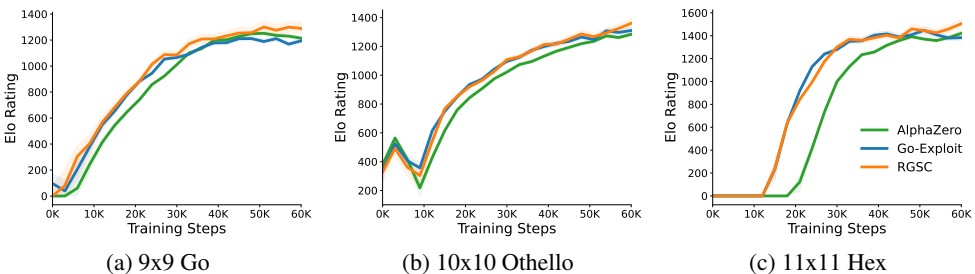

|  |  |  |
|:--:|:--:|:--:|
| (a) 9x9 Go | (b) 10x10 Othello | (c) 11x11 Hex |

Figure 4: Playing performance of AlphaZero, Go-Exploit, and RGSC on three different board games. The shaded area is a 95% confidence interval for the mean.

Figure 4 shows the Elo curves for each method across the three board games. For each game, all models are evaluated against the 150-iteration AlphaZero model, whose Elo rating is fixed at 1000 as the reference point. When comparing the final checkpoint across all methods, RGSC consistently outperforms both baselines in all three games. In 9x9 Go, RGSC surpasses AlphaZero and Go-Exploit by 76 and 96 Elo points, respectively; in 10x10 Othello, the improvements are 70 and 50 Elo points; and in 11x11 Hex, the differences are 84 and 122 Elo points.

Interestingly, we observe that although Go-Exploit achieves a higher Elo than AlphaZero in the early stages of training, its advantage diminishes as training converges. This phenomenon is also evident in the original Go-Exploit experiments. We hypothesize that, during early training, many states exhibit high regret since the model has much to learn, making it easy to select informative states with uniform sampling. As training progresses, however, the number of unfamiliar states decreases, thus uniform sampling becomes less effective. In contrast, RGSC continues to prioritize the remaining high-regret states, allowing it to focus on difficult states and maintain its advantage in the later stages of training.

Table 1: Win rate against established open-source programs on three board games.

|  | AlphaZero | Go-Exploit | RGSC |
|:--|:--:|:--:|:--:|
| 9x9 Go | 45.5%±1.5% | 49.5%±2.0% | **53.6%±2.4%** |
| 10x10 Othello | 51.7%±2.5% | 52.9%±3.3% | **57.8%±3.2%** |
| 11x11 Hex | 83.6%±1.6% | 89.2%±1.8% | **91.1%±2.0%** |

Furthermore, to assess whether the observed improvements remain consistent, we evaluate the final checkpoint by playing against established open-source programs across all three games. We select KataGo (Wu, 2020), one of the strongest open-source Go programs, for 9x9 Go; an alpha-beta search implementation in Ludii (Piette et al., 2020) for 10x10 Othello; and MoHex (Huang et al., 2014), a MCTS-based Hex program that won Computer Olympiad championships, for 11x11 Hex. Detailed settings for each program are listed in subsection A.1. Table 1 summarizes the win rate against these opponents. The results are consistent with the findings in Figure 4, showing that RGSC consistently

outperforms both AlphaZero and Go-Exploit. Overall, these experiments demonstrate that RGSC offers a more efficient search control mechanism, resulting in higher training efficiency and stronger playing performance.

## 4.3 RGSC ON WELL-TRAINED MODELS

Building on the findings in subsection 4.2, where Go-Exploit showed early improvement but failed to yield significant progress as training converged, we now investigate whether RGSC can provide further improvements when starting from an already well-trained model. It is worth noting that mistakes become increasingly rare in such models, making it particularly challenging for the agent to identify and learn from the remaining high-regret states.

To investigate this, we select a large 15-block baseline model trained with the AlphaZero algorithm on 9x9 Go, which required approximately 1,060 NVIDIA RTX A6000 GPU hours and already achieves a strong playing strength. When compared against a KataGo model of the same block size, the baseline achieves a win rate of 69.3%. Similarly, we adopt AlphaZero, Go-Exploit, and RGSC using the same baseline model as the initial weight to ensure a fair comparison. For RGSC, since the original baseline model does not include the regret network, we add the regret network and generate additional self-play games to train it, while keeping the policy and value networks frozen. All three methods are then continued for 40 iterations under identical settings, requiring approximately 100 NVIDIA RTX A6000 GPU hours. Additional training details are provided in subsection A.3.

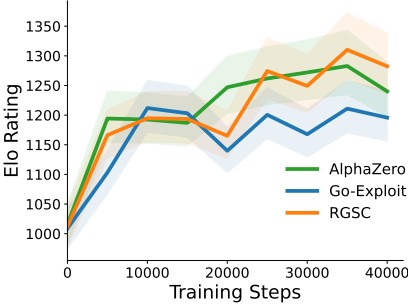

Figure 5: Playing performance of AlphaZero, Go-Exploit, and RGSC when continued from well-trained models. The shaded area is a 95% confidence interval for the mean.

Figure 5 shows the results of continued training. Similar to the findings in subsection 4.2, RGSC achieves the strongest performance, while Go-Exploit performs even worse than AlphaZero. At the final checkpoint, RGSC surpasses AlphaZero by 42 Elo points and Go-Exploit by 87 points. Furthermore, we evaluate the final checkpoint models against KataGo. The original baseline model achieves a win rate of $69.3\% \pm 2.6\%$. After continued training, AlphaZero reaches $70.2\% \pm 2.7\%$ and Go-Exploit $69.2\% \pm 2.7\%$, showing no meaningful improvement. In contrast, RGSC achieves a substantially higher win rate of $78.2\% \pm 2.5\%$, significantly outperforming both baselines. To conclude, these results indicate that RGSC can effectively track remaining self-mistake states even in a well-trained model, thereby achieving further performance improvements.

## 4.4 COMPARISON BETWEEN RANKING AND REGRET IN RGSC

Both the regret value network and regret ranking network can be used to identify candidate states for restarting, but their effectiveness may differ. The regret value network directly estimates regret values, whereas the regret ranking network emphasizes relative ordering. In this subsection, we analyze their differences in identifying high-regret states and examine the impact on search control.

We first train an RGSC variant that relies only on the regret value network for both state selection and regret initialization. Figure 6 presents the training results, showing that the regret ranking network outperforms the regret value network across all three games.

Next, we examine whether the states selected by the two networks indeed correspond to high-regret states. We collect all states from self-play trajectories at each training step and evaluate them by

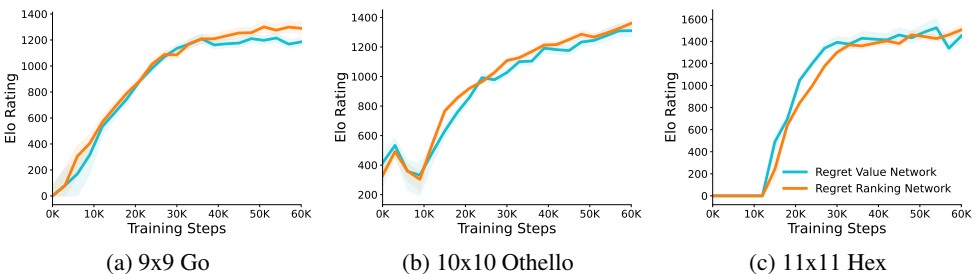

(a) 9x9 Go      (b) 10x10 Othello      (c) 11x11 Hex

Figure 6: Playing performance of RGSC using regret value network and regret ranking network. The shaded area is a 95% confidence interval for the mean.

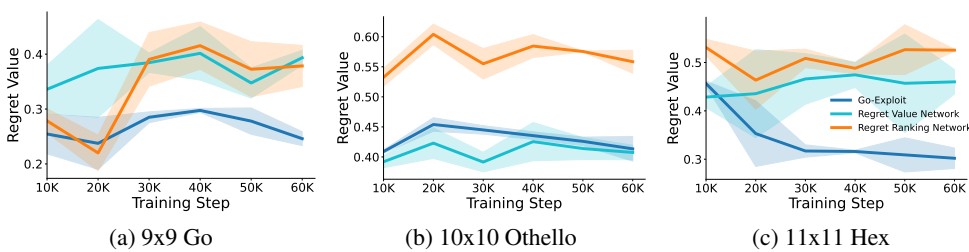

(a) 9x9 Go      (b) 10x10 Othello      (c) 11x11 Hex

Figure 7: The regret values for nodes selected by Go-Exploit, regret value and ranking network. The shaded area is a 95% confidence interval for the mean.

both networks. Since these states come directly from trajectories, their true regret values can be computed according to Equation 2. We then rank the states separately with the regret value and ranking networks, and select the top 2,000 states predicted by each. The true regret values of these selected states are averaged to obtain the average regret of the states identified by each network. Figure 7 shows the average regret of the states selected by each method during training. Overall, the regret ranking network consistently selects states with higher true regret than the regret value network, especially in 10x10 Othello. For convenience, we also include the Go-Exploit approach as a baseline, where the average regret is obtained by randomly sampling 2,000 states. As expected, the uniform sampling approach results in the lowest average regret among all methods. Moreover, the average regret of Go-Exploit decreases during training, especially in Hex, corroborating our hypothesis that Go-Exploit becomes less effective in later stages. In contrast, the regret ranking network maintains a substantially higher average regret even at late training steps, indicating its ability to continually identify difficult states. In summary, these results demonstrate that the ranking objective improves the quality of selected states by prioritizing those with greater learning potential.

### 4.5 Regret Change in Prioritized Regret Buffer

This subsection examines whether the high-regret states in the PRB gradually decrease during training, i.e., whether the model can actually correct its mistakes by repeatedly revisiting those states. Specifically, we record the regret of each state when it first enters the PRB and compare it with its final regret before removal. Figure 8 shows the regret distributions at these two points across all three games. Generally, the distributions consistently shift toward the left (lower regret values) as training progresses. This confirms that states initially associated with high regret are eventually corrected through repeated replay, resulting in reduced regret over time. In addition, by comparing the average regret values, we observe that the average regret decreases significantly across all games: from 0.655 to 0.296 in 9x9 Go, from 0.828 to 0.638 in 10x10 Othello, and from 0.848 to 0.657 in 11x11 Hex. These results demonstrate that RGSC continuously identifies states where the agent struggles, allows them to be self-corrected through repeated revisits until mastered, and then refreshes the buffer with new challenging states. More analyses on high-regret states and the game length of restart states in PRB are provided in Appendix E and Appendix F, respectively.

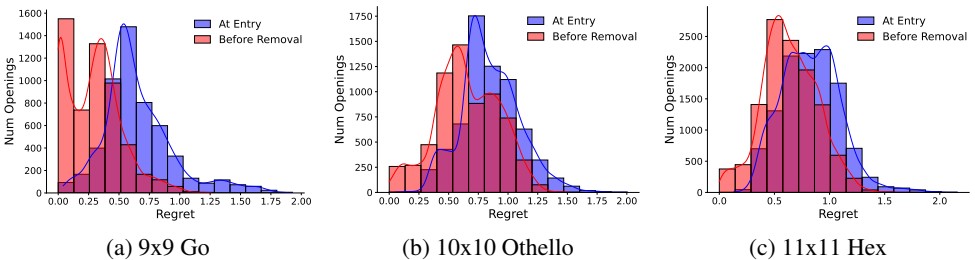

Figure 8: Regret distributions of states in the prioritized regret buffer at entry and before removal.

## 5  DISCUSSION

This paper proposes *Regret-Guided Search Control* (RGSC), an extension of AlphaZero that identifies high-regret states. By integrating a regret network and a prioritized regret buffer, RGSC allows the agent to repeatedly focus on correcting its most critical mistakes, mimicking how humans learn. Experimental results show that RGSC outperforms AlphaZero and Go-Exploit, achieving an average of 77 and 89 Elo, respectively. Furthermore, RGSC successfully improves the win rate against KataGo on a well-trained 9x9 Go model from 69.3% to 78.2%, while both baselines show no improvement. These results demonstrate the learning efficiency and robustness of RGSC.

Although our study focuses on board games, AlphaZero and its successor MuZero (Schrittwieser et al., 2020) are general frameworks, suggesting that RGSC could be applied to more applications beyond games. Specifically, our preliminary experiments (shown in Appendix G) applying RGSC to MuZero on one of the Atari games, *Pac-Man*, show that under the same training budget, RGSC-MuZero reaches 5166 points, compared to 3704 for MuZero. This demonstrates the potential of RGSC to improve learning efficiency beyond board games. Future work can extend RGSC to more domains, such as stochastic environments (Antonoglou et al., 2021) and continuous control tasks (Hubert et al., 2021).

The regret network also provides interpretability by revealing specific weaknesses in the agent's learning. Furthermore, the ability of RGSC to improve even on a well-trained model indicates its scalability to more complex environments such as 19x19 Go or large-scale sequential decision-making problems. We believe RGSC is a promising direction for advancing RL field.

## ETHICS STATEMENT

We do not foresee any ethical issues in this work.

## REPRODUCIBILITY STATEMENT

To reproduce this work, we provided the details of the algorithm in Appendix B, and the hyper-parameters in Appendix A. The source code, trained models used in the experiment, along with a README file and trained models are available at https://rlg.iis.sinica.edu.tw/papers/rgsc/.

## THE USE OF LARGE LANGUAGE MODELS (LLMS)

Large language models (LLMs) were used only for grammar correction and proofreading in the preparation of this paper.

ACKNOWLEDGEMENT

This research is partially supported by the National Science and Technology Council (NSTC) of the Republic of China (Taiwan) under Grant Number NSTC 113-2221-E-001-009-MY3, NSTC 113-2634-F-A49-004, and NSTC 114-2221-E-A49-005.

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

# A EXPERIMENTAL DETAILS

## A.1 TRAINING RGSC IN BOARD GAMES

In this section, we describe the details for training models used in the experiments. The training in section 4 is conducted on a machine with two Intel Xeon Silver 4516Y+ CPUs and four NVIDIA RTX A6000 GPUs. For the implementation of RGSC, the two baseline networks, AlphaZero and Go-Exploit, are implemented based on an open-sourced AlphaZero framework (Wu et al., 2025). During training, 4 self-play workers and 1 optimization worker are used. For RGSC, each worker maintains its own prioritized regret buffer (PRB).

In subsection 4.2, we use hyperparameters shown in Table 2 to train all three methods. For the training of RGSC, the additional hyperparameters are set as follows: the sampling probability $\lambda$ for sampling from the buffer is set to 0.5, the buffer sampling temperature $\tau$ is 0.1, the buffer size is 100, and the EMA coefficient $\alpha$ is 0.5.

Table 2: Hyperparameters for training from scratch (subsection 4.2) and from well-trained model (subsection 4.3).

|  | from scratch | | | from well-trained |
|---|---|---|---|---|
| Parameter | Go | Hex | Othello | Go |
| Board size | 9 | 11 | 10 | 9 |
| Optimizer | SGD | SGD | SGD | SGD |
| Optimizer: learning rate | 0.02 | 0.02 | 0.02 | 0.001 |
| Optimizer: momentum | 0.9 | 0.9 | 0.9 | 0.9 |
| Optimizer: weight decay | 0.0001 | 0.0001 | 0.0001 | 0.0001 |
| MCTS simulation | 200 | 200 | 200 | 400 |
| Softmax temperature | 1 | 1 | 1 | 1 |
| Iteration | 300 | 300 | 300 | 40 |
| Self-Play states per iteration | 160,000 | 120,000 | 120,000 | 160,000 |
| Optimizations per iteration | 200 | 200 | 200 | 1000 |
| Batch size | 1024 | 1024 | 1024 | 256 |
| # Residual blocks | 3 | 3 | 3 | 15 |
| # Residual blocks filters | 256 | 256 | 256 | 128 |
| Replay buffer size | 20 | 20 | 20 | 20 |
| Dirichlet noise ratio | 0.25 | 0.25 | 0.25 | 0.25 |

In subsection 4.3, we train all three methods from already well-trained models with the hyperparameters shown in Table 2. In this training setup, the buffer size is set to 500. To warm up the PRB in RGSC, we generate 1,000 self-play games with a sampling probability $\lambda = 0.5$. These warm-up games are only used in the PRB, and excluded from the training data.

## A.2 COMPUTATIONAL COST ANALYSIS

Although RGSC adds two additional heads, the regret value head and the ranking head, they share the same backbone as the policy and value network, so the additional computation is minimal, especially in larger models. To quantify this, we measured both the neural network inference time and the per-iteration wall-clock time on Go for the 3-block model (used in subsection 4.2) and the 15-block model (used in subsection 4.3). The results are listed in the Table 3. For the 3-block model, RGSC is 1.35x slower in inference and 1.25x slower per iteration compared to AlphaZero or Go-Exploit. However, for the 15-block model, RGSC is approximately 1.03x slower in both inference and per-iteration compared to AlphaZero and Go-Exploit, which are nearly identical. In a realistic setting for past research, AlphaZero used 20 blocks, and KataGo (Wu, 2020) used 20 to 40 blocks, so the overhead becomes negligible. Therefore, RGSC adds almost no extra cost while improving training efficiency in practice.

Table 3: Forwarding time across different blocks of the network.

| # blocks | 3 | 6 | 9 | 12 | 15 | 18 | 20 |
|---|---|---|---|---|---|---|---|
| RGSC | 139.5% | 109.1% | 105.4% | 103.5% | 102.7% | 101.8% | 101.9% |
| AlphaZero | 100.0% | 100.0% | 100.0% | 100.0% | 100.0% | 100.0% | 100.0% |

## A.3 EVALUATION

For evaluating each method, we repeat each experiment twice per game and average the results to minimize stochastic variance. The number of MCTS simulations is set to 400, with the softmax temperature set to 1 for the final action selection.

### A.3.1 EVALUATION AGAINST THE ALPHAZERO

For the experiments in Figure 4 of subsection 4.2, and Figure 6 of subsection 4.3, all methods are evaluated against the AlphaZero baselines every 15 iterations, with 200 games played per evaluation.

### A.3.2 EVALUATION AGAINST OPEN SOURCE PROGRAMS

In subsection 4.2 and subsection 4.3, we evaluated all the methods against open-sourced programs across all three games. The setup of evaluations for each program is outlined below:

**KataGo.** For the evaluation of 9x9 Go in Table 1 of subsection 4.2, we selected KataGo (Wu, 2020) models from ID 1 to 4 listed in Table 4 as baselines. For each baseline, we conduct 200 evaluation games with 100 games as Black and 100 games as White, with the simulation count for KataGo fixed at 400.

For the evaluation against KataGo in subsection 4.3, we pick KataGo with ID 5 in Table 4. In this experiment, actions are selected without applying softmax, and 1,200 evaluation games are conducted for each method.

Table 4: The versions of the selected KataGo models for 9x9 Go.

| ID | Version | # blocks | Elo ratings |
|---|---|---|---|
| 1 | kata1-b6c96-s152505856-d23152636 | 6 | $9833.3 \pm 16.1$ |
| 2 | kata1-b6c96-s165180416-d25130434 | 6 | $9900.6 \pm 16.2$ |
| 3 | kata1-b6c96-s175395328-d26788732 | 6 | $9958.6 \pm 16.9$ |
| 4 | kata1-b10c128-s41138688-d27396855 | 10 | $10138.6 \pm 18.3$ |
| 5 | kata1-b15c192-s86740736-d72259836 | 15 | $11180.1 \pm 16.1$ |

**Alpha-Beta Algorithm in Ludii.** For the evaluation of 10x10 Othello in Table 1 of subsection 4.2, we use the Alpha-Beta algorithm from Ludii (Piette et al., 2020) with search levels set to 2, 3, and 4 as our baselines. For each method in Table 1, a total of 300 games are played, with 150 games as Black and 150 games as White.

**MoHex.** For the evaluation of 11x11 Hex in Table 1 of subsection 4.2, all methods fight against MoHex (Huang et al., 2014). For the MoHex setup, the maximum thinking time is set to 1 second, without using a cache book. Additionally, we use two MoHex baselines with the following search settings: a search width of 15 with a maximum search depth of 5, and a search width of 25 with a maximum search depth of 8.

## B RGSC ALGORITHM

In this section, we describe the details of the RGSC algorithm in Algorithm 1. It contains a value network $f_{regret}$, a regret ranking network $f_{rank}$ within the prioritized regret buffer (PRB). Lines 4–10 specify the procedure of buffer sampling, while Lines 13–23 outline the self-play process, during which all nodes in the search tree are evaluated by the regret network and describe how an opening

$s$ is selected. Line 31–35 describe how an opening $s$ is updated and how it is inserted into the buffer. In this procedure, search tree nodes are inserted into the PRB, enabling us to exploit the search tree while exploring previously unseen states with potentially high regret.

---

**Algorithm 1** RGSC Algorithm

---

**Require:** buffer $\beta$, buffer size $N$, buffer rate $\lambda$, buffer sampling rule $\Psi(\beta)$
**Require:** regret ranking network $f_{rank}$, regret value network $f_{regret}$
 1: Initialize buffer $\beta$
 2: **while** self-play **do**
 3:     Reset the environment to the initial state $s_0$
 4:     Sample random number $p \in (0, 1)$
 5:     **if** $p < \lambda$ **then**
 6:         Sample a opening $o$ from $\beta$ with $\Psi(\beta)$
 7:         Set $o$ as the starting state
 8:     **else**
 9:         Set $s_0$ as the starting state
10:     **end if**
11:     Initialize global candidate state $s' \leftarrow \emptyset$, ranking score $\gamma' \leftarrow -\infty$, regret value $g' \leftarrow -\infty$
12:     Self-play from the starting state
13:     **while** the environment state $s_t$ is not terminal **do**
14:         Run MCTS on current state $s_t$ and select action $a$
15:         **for** each search node $s$ in the MCTS tree **do**
16:             Predict ranking score $\gamma_s$
17:             Predict regret value $g_s$
18:             **if** $\gamma_s > \gamma'$ **then**
19:                 $s' \leftarrow s$;   $\gamma' \leftarrow \gamma_s$;   $g' \leftarrow g_s$
20:             **end if**
21:         **end for**
22:         Execute $a$, observe next state $s_{t+1}$, set $t \leftarrow t + 1$, $s_t \leftarrow s_{t+1}$
23:     **end while**
24:     **for** each $s_t$ in the self-play trajectory $\{s_n, s_{n+1}, ..., s_T\}$ **do**
25:         Use final return to compute regret $\mathcal{R}(s_t)$
26:         Get ranking score $\gamma_{s_t}$
27:         **if** $\gamma_{s_t} > \gamma'$ **then**
28:             $s' \leftarrow s_t$;   $\gamma' \leftarrow \gamma_{s_t}$;   $g' \leftarrow \mathcal{R}(s_t)$
29:         **end if**
30:     **end for**
31:     **if** $p < \lambda$ **then**
32:         Update the regret of opening $o$ in buffer $\beta$ using the EMA rule in Equation 13
33:     **else**
34:         Store the candidate $s'$ with its regret $g'$ into buffer $\beta$
35:     **end if**
36: **end while**

---

## C   Detail for toy model experiment

In the experiments in subsection 4.1, we implemented a simple Q-learning example on sparse-reward binary trees with five and six levels. In Q-learning, the learning rate is set to 0.1, $\epsilon$ is set to 0.1 in epsilon greedy, and the discount factor $\gamma$ is also set to 0.1. We compare the training speed of three opening sampling strategies for selecting the starting state in self-play. The first one always starts from the root. The second one uniformly samples a non-terminal state from a fixed-size first-in-first-out buffer that stores past trajectories with a buffer rate of 0.5, similar to the GEVC method in Go-Exploit. The third strategy, the regret-based method, samples states in proportion to their regret values. For the random method and regret method, a buffer rate of 0.5 is applied. Each method is trained for 6,000 iterations. For every 100 iterations, 6000 games are played for evaluation, and the average reward is calculated. To reduce the variance, we use 25 different seeds in the experiments.

In Figure 3b, we compare the difference between the root's estimated Q-value and its theoretical optimal Q-value across the three methods.

## D  PERFORMANCE UNDER DIFFERENT HYPERPARAMETER SETTINGS IN RGSC

In this section, we explore the hyperparameters in RGSC, including the sampling probability ($\lambda$), the buffer sampling temperature ($\tau$), the buffer size ($\kappa$), and the EMA coefficient ($\alpha$) across 9x9 Go, 10x10 Othello, and 11x11 Hex. In the ablation study, we aim to choose the setting with less computational cost and better performance. Additionally, we select the setting that demonstrates stable and consistent results across different games.

### D.1  SAMPLING PROBABILITY

We evaluate different sampling probabilities ($\lambda$), which represent the probability (as introduced in Section 3.3) of starting a self-play trajectory from a state sampled from the PRB. We evaluate $\lambda \in \{0.2, 0.5, 0.9\}$, as shown in Figure 9. Overall, RGSC with $\lambda = 0.5$ performs consistently well across the three games, so we use $\lambda = 0.5$ for our final setting. In 11x11 Hex (Figure 9c), a high sampling probability ($\lambda = 0.9$) yields an early improvement around 10K training steps, but subsequently causes significant performance instability toward the end of the training process. Moreover, in 9x9 Go, it also shows that $\lambda = 0.9$ exhibits marginally reduced stability (Figure 9a). These may be due to overly frequent sampling of recent self-play data from the PRB.

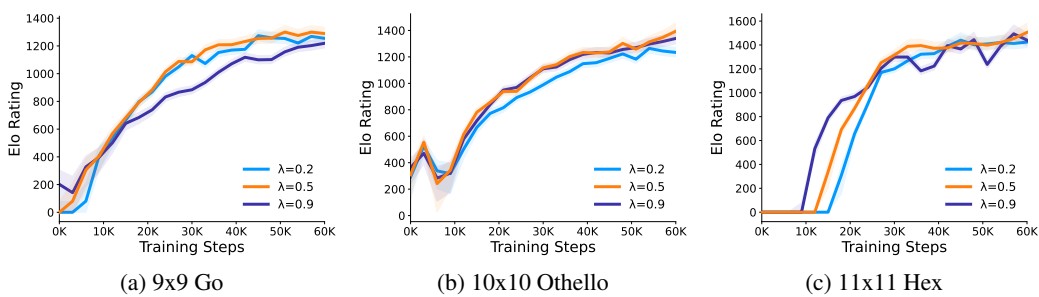

(a) 9x9 Go            (b) 10x10 Othello            (c) 11x11 Hex

Figure 9: Playing performance of different sampling probabilities ($\lambda$) across three games. The orange curve is the final RGSC setting.

### D.2  BUFFER SAMPLING TEMPERATURE

Furthermore, we investigate the effect of the buffer sampling temperature ($\tau$) described in subsection 3.3, where a higher value leads to a more uniform softmax distribution for the PRB. We test $\tau \in \{0.1, 0.5, 1\}$, as shown in Figure 10, and the results show that $\tau = 0.1$ achieves the best performance.

### D.3  BUFFER SIZE

We evaluate different buffer sizes, $\kappa \in \{100, 500, 1000\}$ in RGSC, as shown in Figure 11. The results show no significant difference in performance, so we use $\kappa = 100$ in our final setting to minimize computational cost.

### D.4  EMA COEFFICIENT

For the EMA coefficient used to update the regret values in Equation 13 of the main text, we test $\alpha \in \{0.1, 0.5, 1\}$, as shown in Figure 12, finding that $\alpha = 0.5$ generally yields the best performance in RGSC. Note that $\alpha = 1$ only uses the regret calculated from the newly finished self-play game,

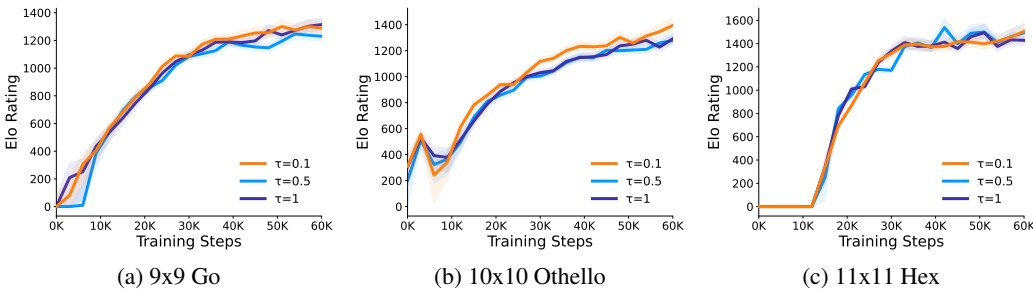

Figure 10: Playing performance of different buffer sampling temperatures ($\tau$) across three games. The orange curve is the final RGSC setting.

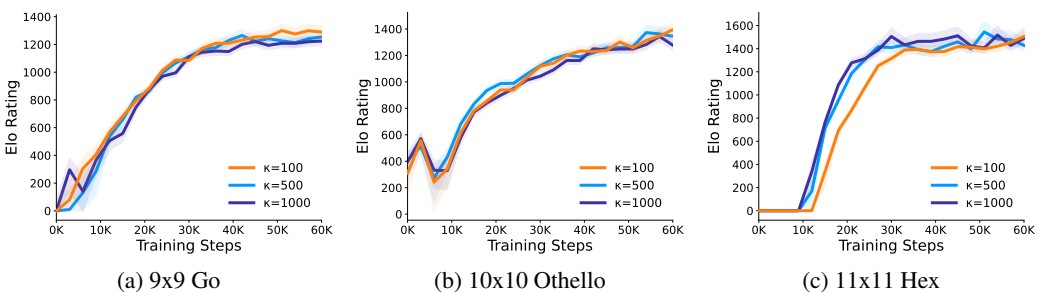

Figure 11: Playing performance of different buffer sizes ($\kappa$) across three games. The orange curve is the final RGSC setting.

explaining its slightly unstable performance in 10x10 Othello (Figure 12b) and 11x11 Hex (Figure 12c).

In conclusion, these experiments show that RGSC is not highly sensitive to hyperparameter choices and that our recommended configuration works well across different games, demonstrating the overall robustness of the method.

## E  ANALYSIS OF HIGH-REGRET STATES

To further examine the high-regret states stored in the regret buffer, we analyze several examples that RGSC frequently selects in 9x9 Go, 10x10 Othello, and 11x11 Hex. In Figure 13, we show examples of high-regret states sampled from the regret buffer of RGSC in these three games. To evaluate whether the agent has learned the technique for solving these high-regret states, we evaluate the RGSC agents against an AlphaZero baseline, using this state as the starting position, as shown in Figure 14. Finally, we track the evolution of regret values and corresponding game outcomes for these examples throughout training, showing their relationship in Figure 15.

**9x9 Go.** In Figure 13a, White can only win by keeping the stones at I8 (A) alive. Once White plays at H9 (B), a seki (Niu et al., 2006) arises at the top-right corner. In a seki, neither player can capture the other's stones—the first player to play inside the seki area will have their stones captured. Thus, forming the seki is the only path to victory for White. As shown in Figure 14a, the win rate for this example increases from 47% to over 90% within a few training iterations, indicating that RGSC successfully learns the strategy required to win from this high-regret state.

To further verify whether the agent has truly mastered this high-regret situation, we evaluate a 9x9 Go state (Figure 13b) that shares the same structural pattern as the one in Figure 13a, where White can only win by playing at H9 (C). In Figure 14a, the agent's win rate in this state increases from 47% to 85%. This shows that RGSC enables the agent to identify weaknesses in its current policy and correct them automatically, and generalize its learned strategies to structurally similar situations.

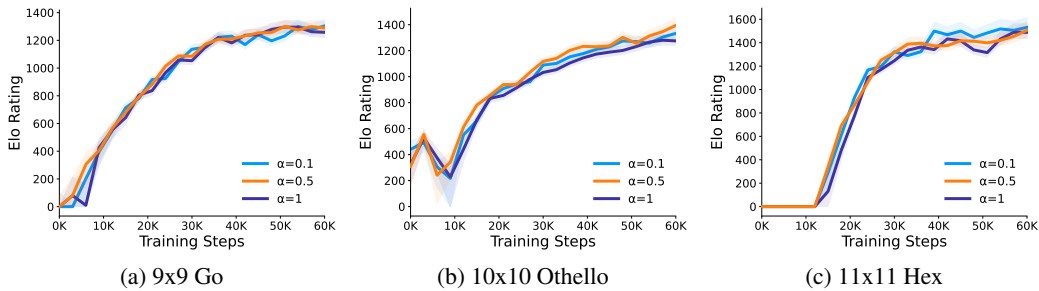

(a) 9x9 Go  (b) 10x10 Othello  (c) 11x11 Hex

Figure 12: Playing performance of different EMA coefficients ($\alpha$) across three games. The orange curve is the final RGSC setting.

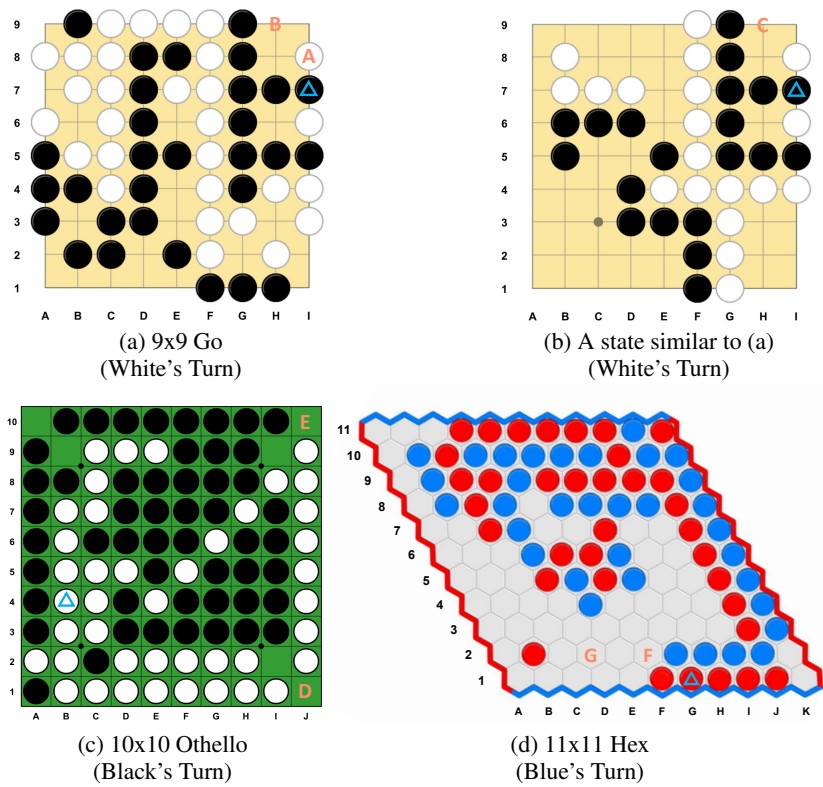

(a) 9x9 Go
(White's Turn)

(b) A state similar to (a)
(White's Turn)

(c) 10x10 Othello
(Black's Turn)

(d) 11x11 Hex
(Blue's Turn)

Figure 13: High-regret state examples. Blue triangles indicate the last moves.

**10x10 Othello.** A high-regret example of 10x10 Othello is shown in Figure 13c, where Black secures a guaranteed win by playing J1 (D) followed by J10 (E). At 13K training steps, the agent's win rate for Black at this example is 82%, as shown in Figure 14b. After the first update, no significant improvement was observed. However, following the second update at 13.8K training steps, the win rate sharply increases to 100%. This indicates that RGSC has successfully learned the correct strategy for this high-regret state and no longer makes mistakes when encountering it.

**11x11 Hex.** Finally, a high-regret example of 11x11 Hex is shown in Figure 13d. In this example, Blue can win by playing F2 first (F), followed by D2 (G). After the first update at 18.4K training steps, the win rate for this example surges to 88%, as shown in Figure 14c. After an additional round of training at 19.0K steps, the win rate further increases to 91%, showing a 14% improvement within a single iteration. These results demonstrate that RGSC not only enables the agent to correct

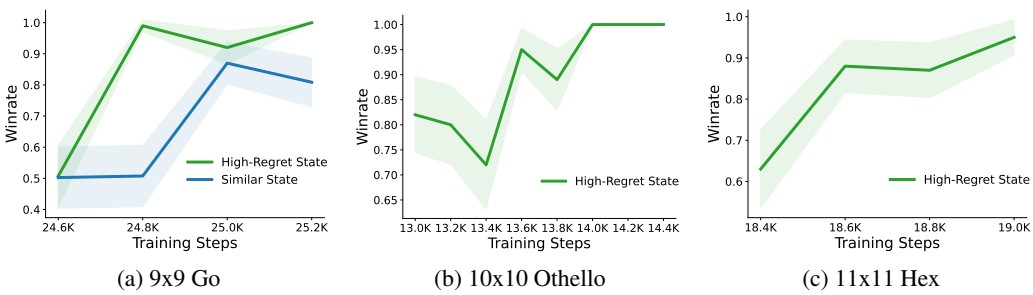

Figure 14: Win rates of high-regret states during RGSC training.

its previous mistakes but also enhances the interpretability of the AlphaZero framework by revealing how the model progressively learns to resolve high-regret states.

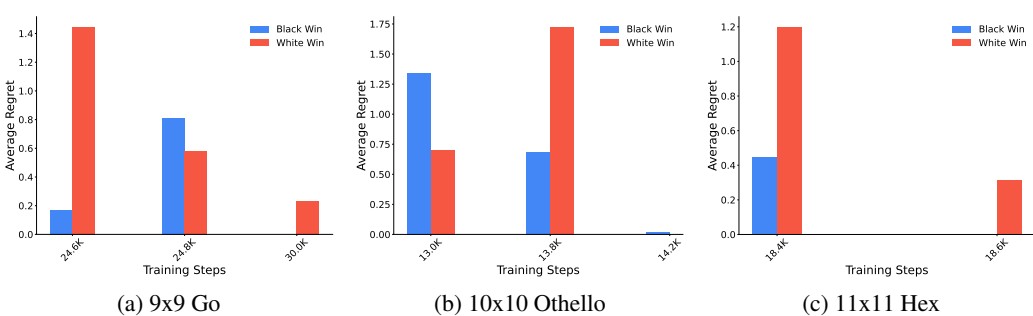

Figure 15: Relationship between game outcomes and regrets during training.

Finally, we look into the relationship between the game outcomes and the corresponding regrets of these examples throughout the training, as shown in Figure 15. For a state where White always wins under the optimal play, the regret associated with White's victory decreases as the agent's predictions on that state become more accurate. We first examine the examples in 9x9 Go and 11x11 Hex, where White (represented as Red in Hex) can win under the optimal policy. As shown in Figure 15a and Figure 15c, the regret for White's win diminishes with more training iterations on these openings, while the regret for Black's win increases correspondingly. On the other hand, we examine the example in 10x10 Othello, where Black has a guaranteed win, as shown in Figure 15b. The average regret values associated with Black's wins decrease as training progresses. By 14.2K training steps, no outcomes of White wins are observed, showing that the agent has fully learned the correct strategy to win in this example. In conclusion, the convergence of regret indicates that the agent is progressing toward the optimal policy during training.

## F ANALYSIS OF PRIORITIZED REGRET BUFFER

In this section, we investigate the openings' attributes in PRB. In subsection F.1, we investigate the distribution of opening lengths during training and how the agent adapts to different phases of the game. Next, in subsection F.2, we track how the regret values of the openings evolve throughout training, showing that the agent becomes increasingly familiar with high-regret states and progressively refines its policy.

### F.1 OPENING-LENGTH DISTRIBUTIONS IN TRAINING PROCESS

In board games such as Go, Hex, and Othello, the game is typically divided into three phases: early game, midgame, and endgame. Among these, the midgame typically involves the most complex tactics and combinatorial challenges, offering the greatest potential for learning. To see which phase

of the game is preferred for learning during training, we examine the length of openings used in PRB throughout the training process. In Figure 16, Figure 17, and Figure 18, we compare RGSC with a variant that excludes the regret ranking network and selects openings solely from trajectories based on computed regret.

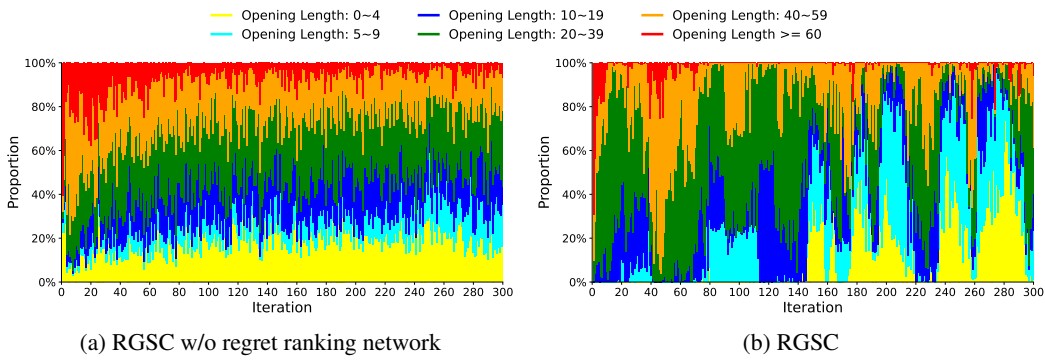

(a) RGSC w/o regret ranking network      (b) RGSC

Figure 16: Change in the proportion of openings with different lengths across training in 9×9 Go.

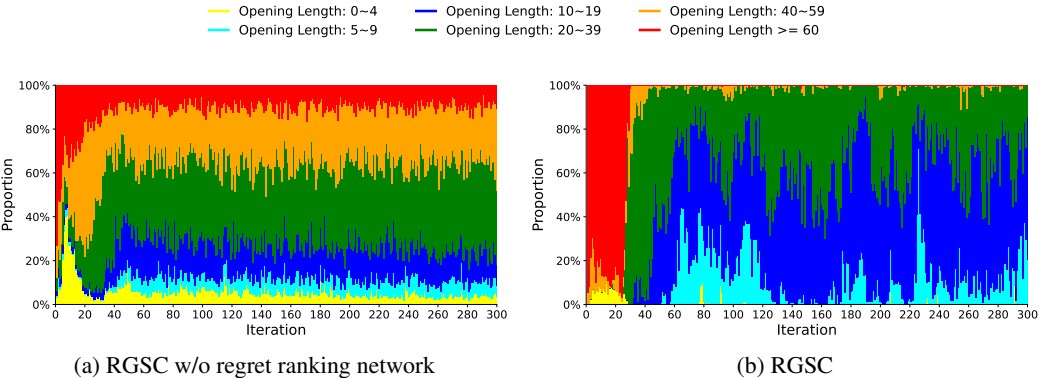

(a) RGSC w/o regret ranking network      (b) RGSC

Figure 17: Change in the proportion of openings with different lengths across training in 10×10 Othello.

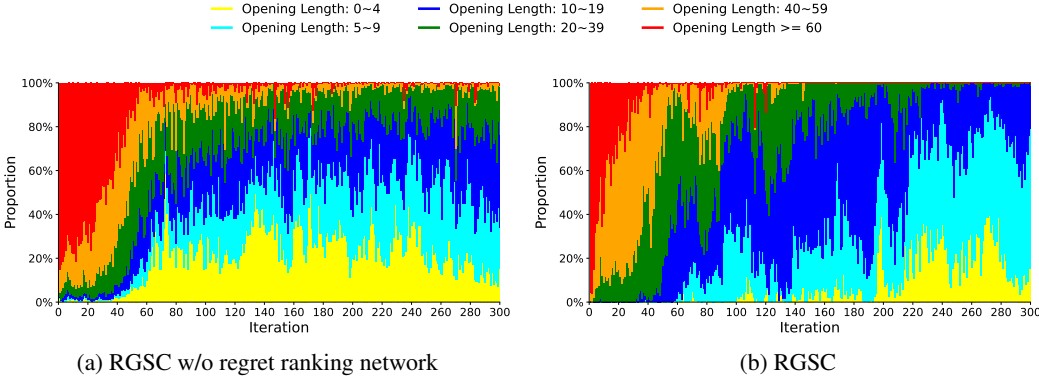

(a) RGSC w/o regret ranking network      (b) RGSC

Figure 18: Change in the proportion of openings with different lengths across training in 11×11 Hex.

Figure 16 shows the dynamics of the proportion of openings with different lengths throughout the training process in RGSC with and without a regret ranking network in 9x9 Go. RGSC without regret ranking network shows no significant preference for opening lengths, as shown in Figure 16a. In contrast, RGSC exhibits a clear phase-aware curriculum. In the early stages of training, it favors openings with opening lengths ranging from 20 to 39, corresponding to midgame positions,

which are the most complex and informative. As training progresses and the agent's overall strength improves, so that games reach midgame phases earlier, the distribution shifts toward shorter opening lengths, indicating that RGSC continues to target the most exploration-worthy states as the MCTS value estimation improves.

Figure 17 and Figure 18 show the results of the analysis in Hex and Othello. We observe the same qualitative pattern: relative to the RGSC without a regret ranking network, RGSC progressively biases toward shorter opening lengths as training advances. The consistency across various games indicates that RGSC selects openings based on the model's playing strength, allowing it to identify the most valuable learning states at different training stages.

The dynamics of opening lengths show that RGSC induces an implicit, data-driven curriculum on game phases: initially focusing on complex midgame positions, then shifting to shorter openings as the model's playing strength improves. In contrast, RGSC without regret ranking network remains less sensitive to the phases and difficulty of the game.

## F.2 DECREASING REGRET VALUES ACROSS TRAINING

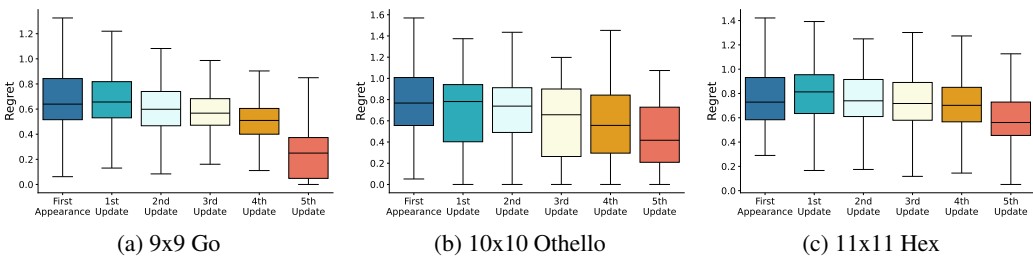

(a) 9x9 Go  (b) 10x10 Othello  (c) 11x11 Hex

Figure 19: Changes in the values of openings from their first appearance to the final update during training.

We analyze how the regret values of openings stored in the regret buffer evolve during training. Specifically, we track the regret values of openings from their first appearance to their final removal, capturing their updates across five training iterations. As shown in Figure 19, the regret values consistently decrease with each update, indicating that the model becomes increasingly familiar with these high-regret openings. By the final update, the overall regret distribution shifts toward lower values, resulting in the lowest regret levels observed across all iterations. This demonstrates that the agent progressively focuses on challenging states and refines its policy to reduce the associated regret over time.

## G RGSC ON ATARI GAMES

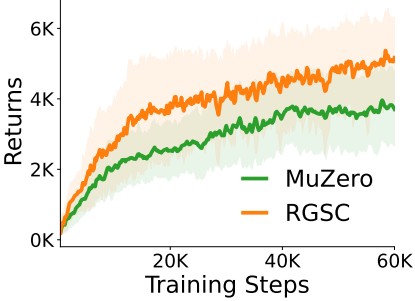

Figure 20: Playing performance of MuZero, and RGSC on Ms. Pac-Man. The shaded area is a 95% confidence interval for the mean.

RGSC can also be generalized to MuZero (Schrittwieser et al., 2020) or other AlphaZero-variants. To demonstrate this, we further integrate RGSC into MuZero in a large-scale domain, the Atari benchmark. Specifically, we select one of the Atari games, *Ms. Pac-Man*, in our evaluation. Unlike board games, where the outcome is only available at the terminal state, we use the internal rewards to compute n-step return for $z$ in Equation 2 for all intermediate states in Atari games.

Since Go-Exploit is implemented only within AlphaZero for board games, our experiment focuses on comparing RGSC with MuZero. Throughout the entire training process, RGSC significantly outperforms MuZero, as shown in Figure 20. At the final iteration, RGSC achieves an average score of 5166 points, while MuZero only achieves 3704 points, demonstrating the generality of RGSC when applied to other AlphaZero-like style algorithms and highlighting its ability to handle complex tasks.

