# OpenReview forum: "Regret-Guided Search Control for Efficient Learning in AlphaZero"
_ICLR.cc/2026/Conference — ICLR 2026 Poster_

### Official Review · Reviewer_WKT8 · 2025-10-29

**Soundness:** 2
**Presentation:** 2
**Contribution:** 2
**Rating:** 2
**Confidence:** 3

**Summary:**

The paper proposes a new technique for improving the efficiency of training of MCTS agents by choosing which states to start playing from, rather than always starting from the very beginning (AlphaZero) or from uniform sampling of visited states (Go-Exploit).

The authors call this method Regret-Guided Search Control (RGSC). It prioritizes sampling states that have high regret, where regret is the discrepancy between the agent's evaluation and the game's actual outcome. To predict such regrets, they train both a regret value network and a regret ranking network. The predictions are then used to select states from a prioritized regret buffer (PRB).

The authors evaluate on a number of games, including Go, Othello, and Hex. They do an ablation study to see whether the regret ranking network actually helps. They also measure whether high-regret states in the PRB gradually decrease during training.

**Strengths:**

This paper presents a seemingly promising idea of choosing which states to play from, rather than always starting from the very beginning (AlphaZero) or from uniform sampling of visited states (Go-Exploit). Conceptually, it makes a lot of sense that doing so could vastly increase the efficiency of the learning process.

**Weaknesses:**

According to the plots, the experimental results do not show a strong advantage of the proposed method in terms of Elo rating.

Figure 4 does not show a strong advantage of RGSC over Go-Exploit in terms of Elo rating.

Figure 5 does not show a strong advantage of RGSC over AlphaZero in terms of Elo rating.

Figure 6 does not show a clear superiority of the regret ranking network over the regret value network, especially in 11 x 11 Hex. In the last case, the former *underperforms* the latter for most of the run.

Also, some choices in the design of the ranking surrogate objective have an unclear justification (more comments on this below). It would greatly help if the authors justified each step of their design, as well as how the hyperparameters would be chosen in practice.

Page-by-page corrections:

Page 1:

learning efficient -> learning-efficient

Page 2:

using starting states -> choosing starting states

later training -> later stages of training

Page 5:

redundant parentheses around R(s_j)^(1/τ) in the denominator for the PRB expression

Page 6:

Figures 3, 4, 5, 6: "The shaded area is the error bar with 95% confidence interval" -> "The shaded area is a 95% confidence interval for the mean", if this is what you meant.

**Questions:**

Page 1:

"represented as cells" What does this mean?

Page 4:

A ranking-based objective might partially address non-stationarity, but can't the ranking (relative order) also change with further training?

Page 5:

Why does the exponential transformation aid optimization? If the aim is non-negativity, why not use softplus or squareplus, which are more stable and less susceptible to overflow for large values?

How do you choose the sampling temperature τ, the EMA rate α, and the probability λ in a principled way in practice?

Appendix 1 describes the hyperparameter values that were chosen. Were these obtained via trial-and-error? How sensitive is the performance to these choices?

The design of outputting the ranking score and getting the resulting restart distribution via a softmax seems a little ad hoc. Can you explain this?

How many trials were used for each figure (3, 4, 5, 6, 7) and table (1, 2)? Were these independent runs of the whole training process?

Page 6:

"All methods use a 3-block residual network" What activation function and weight initialization (presumably He) was used?

Page 8:

Figure 7 lacks uncertainty bands. Was this run for multiple trials?

---

> ### Author Response · Authors · 2025-11-22
> **Response to Reviewer WKT8 (1/2)**
>
> Thank you for your comments and for providing insightful feedback. We address each weakness and question below.
>
> > According to the plots, the experimental results do not show a strong advantage of the proposed method in terms of Elo rating.
>
> Regarding Figure 4, we agree that RGSC and Go-Exploit show similar Elo curves in the early stages, as already noted in line 319 of the main text. However, as training progresses, RGSC becomes more effective than Go-Exploit, as shown in the later stages of Figure 4 and in the experiments using well-trained models in Figure 5.
>
> Regarding Figure 5, RGSC still achieves better performance than AlphaZero. More importantly, when evaluated against a strong external program (KataGo), RGSC significantly improves the win rate from 70.2% for AlphaZero and 69.2% for Go-Exploit to 78.2%.
>
> Overall, Figures 4 and 5 show that RGSC achieves better performance in multiple settings (early training, late training, and external evaluation), offering a simple yet consistently effective improvement over AlphaZero and Go-Exploit.
>
> Regarding Figure 6, we acknowledge that the difference between the regret value and regret ranking networks is smaller in the 11×11 Hex. However, both methods are introduced in this paper. Across all three games, the ranking network performs well and provides a single, stable design that generalizes across game types, which we believe highlights the robustness of RGSC.
>
> > Page-by-page corrections:
>
> Thank you for pointing these out. We have revised each accordingly in the revision.
>
> > Page 1: "represented as cells" What does this mean?”
>
> The visited promising states are referred to as "cells" in Go-Explore. Since this term is not used in the rest of our paper, we removed the phrase "represented as cells" from the original sentence in the revision to avoid confusion. Thank you for pointing this out.
>
> > Page 4: A ranking-based objective might partially address non-stationarity, but can't the ranking (relative order) also change with further training?
>
> Both the regret-based objective and the ranking-based objective are inherently non-stationary, since the targets change during training. However, the ranking-based objective partially reduces the impact of non-stationarity because it only requires learning relative order rather than their absolute regret values. For example, if three states originally have regrets [4, 2, 1] and later become [2, 1, 0], a value-based regret objective must learn all three targets, whereas a ranking-based objective does not need to update their relative order at all. This makes learning easier, especially under heavily imbalanced regret distributions, and provides a more stable training signal overall.
>
> > Page 5: Why does the exponential transformation aid optimization? … why not use softplus or squareplus …
>
> The exponential transformation is not for non-negativity, but to amplify the relative differences among regret values, which strengthens the ranking signal. In principle, any strictly increasing function (e.g., softplus or squareplus) could be used; exponential scaling is simply more effective at separating high-regret states while remaining compatible with the log-sum-exp structure in Equation (6). In our early design, the exponential version produced a more stable ranking signal than variants without it, so we adopted it in RGSC.

---

> ### Author Response · Authors · 2025-11-22
> **Response to Reviewer WKT8 (2/2)**
>
> > Page 5: How do you choose the sampling temperature $\tau$, the EMA rate $\alpha$, and the probability $\lambda$ in a principled way in practice?
>
> > Page 5: Appendix 1 describes the hyperparameter values that were chosen. Were these obtained via trial-and-error? How sensitive is the performance to these choices?
>
> Please refer to our general response on "Regarding the choice of hyperparameters in RGSC".  Most hyperparameters shown in Table 2 are inherited directly from the original AlphaZero algorithm and were directly taken from prior works.
>
> > Page 5: The design of outputting the ranking score and getting the resulting restart distribution via a softmax seems a little ad hoc. Can you explain this?
>
> Using softmax in equation 3 is not ad hoc. Since the ranking score is an unnormalized score, using a softmax is a common way to convert such scores into a probability distribution in ranking models such as the Bradley-Terry and Plackett-Luce. It preserves the relative ordering while naturally emphasizing higher-ranked states, making it a suitable choice for deriving the restart distribution.
>
> > Page 5: How many trials were used for each figure (3, 4, 5, 6, 7) and table (1, 2)? Were these independent runs of the whole training process?
>
> For Figures 3, 4, 6, and 7, we repeat each experiment twice per game and average the results to reduce variance. These two runs are independent full training runs. We have included this information in Appendix A.2. For Figure 5, we provide only one run due to the high computational cost.
>
> > Page 6: "All methods use a 3-block residual network" What activation function and weight initialization (presumably He) was used?
>
> For settings not related to RGSC, we simply follow the common settings used in AlphaZero. The activation function is ReLU, and the weight initialization follows the standard He initialization.
>
> > Page 8: Figure 7 lacks uncertainty bands. Was this run for multiple trials?
>
> Figure 7 was originally based on a single run. We have updated Figure 7 to include two independent runs along with the corresponding uncertainty bands. The overall trends remain consistent: both the value and ranking outperform Go-Exploit, and the ranking performs the best overall. Thank you for pointing this out.
>
> We hope the added experiments and clarifications address your concerns. Our experiments show a strong advantage that RGSC outperforms under either early or late training, and we have included detailed experiments for different hyperparameters. These results show that RGSC is robust and broadly applicable (we have also applied RGSC to Atari games in Appendix H), representing a meaningful contribution. We believe our work is valuable and deserves a higher evaluation score and sincerely hope the reviewer will reconsider the evaluation based on the clarifications we have provided.

---

> ### Comment · Area_Chair_HuZx · 2025-11-26
>
> The evaluation of this submission has a large variance so we need to gather more information from both the authors and the reviewers. Please take a look at the author's response and see how it affects your evaluation.
>
> Best,
> AC

---

> > ### Author Response · Authors · 2025-11-27
> >
> > Dear reviewer WKT8,
> >
> > Thank you for your review and valuable feedback. We have provided detailed responses and additional experiments. As the discussion phase is ending, we would greatly appreciate it if you could review our rebuttal. We are willing to address any further concerns during the remaining rebuttal period. Thank you again for your time and consideration.
> >
> > Sincerely,
> >
> > Authors

---

### Official Review · Reviewer_4XJP · 2025-11-01

**Soundness:** 3
**Presentation:** 3
**Contribution:** 3
**Rating:** 6
**Confidence:** 3

**Summary:**

This paper proposes Regret-Guided Search Control (RGSC) to improve the efficiency of AlphaZero-style reinforcement learning. RGSC uses a regret network to identify high-regret states, positions where the agent’s evaluation most diverges from the outcome, and stores them in a prioritized buffer for replay as new starting positions. Experiments on 9×9 Go, 10×10 Othello, and 11×11 Hex show RGSC outperforms AlphaZero and Go-Exploit by 77 and 89 Elo points on average, and it improves win rates even from strong pre-trained models. Overall, RGSC provides an effective mechanism for search control, enhancing both the efficiency and robustness of AlphaZero training.

**Strengths:**

- Novel and well-motivated idea: The use of regret-guided search control is an intuitive and principled extension of AlphaZero that better mimics human-style targeted learning.
- Clear technical contribution: Introducing a regret network and a ranking-based objective to identify and prioritize high-regret states is a concrete and original addition to the AlphaZero framework.
- Strong empirical results: RGSC shows consistent performance improvements across multiple domains (Go, Othello, Hex) and against strong baselines (AlphaZero, Go-Exploit).
- Robustness to training stage: The method continues to yield improvements even when applied to well-trained models, suggesting it generalizes beyond early training efficiency.

**Weaknesses:**

- Limited theoretical justification: The paper mainly relies on empirical validation; it lacks a clear theoretical analysis of why regret-based prioritization should improve sample efficiency.
- Scalability concerns: Results are shown only on small board sizes (9×9 Go, 10×10 Othello, 11×11 Hex). It’s unclear if RGSC scales to larger or more complex domains (e.g., 19×19 Go).
- Computational overhead: Maintaining a regret network and prioritized buffer might add non-trivial computational cost; efficiency trade-offs are not clearly discussed.
- Ablation detail: The paper would benefit from more ablation or sensitivity analysis (e.g., effect of buffer size, ranking loss design, or sampling strategy).

**Questions:**

- Would RGSC remain effective on large-scale domains like 19×19 Go or Atari, where regret estimation might be noisier?
- How significant is the computational overhead introduced by the regret network compared to standard AlphaZero training?
- Could the regret signal be estimated directly from existing AlphaZero components (e.g., policy/value mismatch) without training a separate network?
- How sensitive is RGSC’s performance to the design of the ranking-based loss or the regret buffer prioritization mechanism?

---

> ### Author Response · Authors · 2025-11-22
> **Response to Reviewer 4XJP**
>
> Thank you for your positive comments and for providing insightful feedback. We address each weakness and question below.
>
> > W1: Limited theoretical justification
>
> The idea of regret-based prioritization is intuitive: these are precisely the states the model struggles with, and revisiting them naturally improves learning efficiency. Most prior work on search control, including Go-Explore and Go-Exploit, relies mainly on empirical results without theoretical analysis or guarantees. Our empirical experiments similarly demonstrate the effectiveness of RGSC. We agree that a theoretical analysis would further strengthen our method, and we consider this a promising direction for future work.
>
> > W2 & Q1: on large-scale domains like 19×19 Go or Atari
>
> Please refer to our general response on "Regarding applying RGSC to other environments beyond board games".
>
> > W3 & Q2: Computational overhead
>
> Please refer to our general response on "Regarding the computational cost of RGSC".
>
> > W4 & Q4: ablation or sensitivity analysis (e.g., effect of buffer size, ranking loss design, or sampling strategy).
>
> Please refer to our general response on "Regarding the choice of hyperparameters in RGSC".
>
> > Q3: Could the regret signal be estimated directly from existing AlphaZero components (e.g., policy/value mismatch) without training a separate network?
>
> In AlphaZero, only the trajectory states (e.g., root node) have training labels (MCTS search distribution for policy and win/lose for value), and thus only these states can compute a policy/value mismatch. To leverage a broader set of states, especially the many nodes inside the MCTS tree, it is necessary to train a network that can predict regret for all states. This is why we choose to train a separate network.
>
> Thank you again for your positive comments. We hope our additional experiments and clarifications address your concerns.

---

> > ### Comment · Reviewer_4XJP · 2025-11-23
> >
> > Thank you for the clarifications. The explanations are satisfactory, and I remain positive about the work. I will keep the score I originally gave.

---

> > > ### Author Response · Authors · 2025-11-25
> > >
> > > Thank you for your kind response and support! If there is anything further we can do, please let us know. We are willing to engage in any further discussions or conduct additional experiments as needed during the rebuttal period. Thank you again for your time and effort.

---

### Official Review · Reviewer_taWz · 2025-11-01

**Soundness:** 2
**Presentation:** 3
**Contribution:** 2
**Rating:** 2
**Confidence:** 2

**Summary:**

The paper proposes the RGSC, which introduces a regret network to identify high-regret states where the agent's evaluation significantly deviates from the actual game outcome. And experiments are conducted on 9×9 Go, 10×10 Othello, and 11×11 Hex.

**Strengths:**

1. The paper defines “regret” as the average cumulative deviation from the current state to terminal states. This formulation captures the long-term impact of mistakes while avoiding the locality limitations of single-step error measures.

2. The method is evaluated on several board-game domains, with a reasonably comprehensive experimental suite within that problem class.

**Weaknesses:**

well RGSC generalizes beyond these discrete, perfect-information domains.

2. Dependence on terminal observability: RGSC’s regret computation requires access to complete state-to-terminal outcomes. In many real-world or continuous-control tasks (e.g., robot, Atari), the notion of a single terminal outcome can be ambiguous or trajectories cannot be fully recovered, making the regret signal difficult or impossible to compute reliably.

3. Baselines: the baselines appear outdated (2023 as the most recent); stronger, more recent baselines should be included to better position RGSC relative to the current state of the art.

4. Hyperparameter ablation: the method involves many hyperparameters, but the paper lacks ablation studies to show sensitivity and justify the chosen settings.

**Questions:**

1. Can RGSC be generalized to other benchmarks such as Atari or robotic control tasks where terminal outcomes are ambiguous or partial observability is present?

2. The method uses many hyperparameters. Have these been validated via ablation studies?

---

> ### Author Response · Authors · 2025-11-22
> **Response to Reviewer taWz**
>
> Thank you for your comments and for providing insightful feedback. We address each weakness and question below.
>
> > W1 & W2 & Q1: generalized to other benchmarks
>
> Please refer to our general response on "Regarding applying RGSC to other environments beyond board games".
>
> > W3: Baselines: the baselines appear outdated (2023 as the most recent); stronger, more recent baselines should be included to better position RGSC relative to the current state of the art.
>
> We respectfully disagree that our baselines are outdated. To the best of our knowledge, Go-Exploit is the most recent and most relevant method that explicitly studies search control for AlphaZero. At the time of submission, we conducted a thorough survey of the literature and found no subsequent work proposing a newer or stronger search-control mechanism for AlphaZero training. If the reviewer is aware of a more recent or stronger baseline, we would greatly appreciate concrete references.
>
> > W4 & Q2: Hyperparameter ablation
>
> Please refer to our general response on "Regarding the choice of hyperparameters in RGSC".
>
> We hope the added experiments and clarifications address your concerns. These results show that RGSC is robust and broadly applicable, representing a meaningful contribution. We believe our work is valuable and deserves a higher evaluation score and sincerely hope the reviewer will reconsider the evaluation based on the clarifications we have provided.

---

> ### Comment · Area_Chair_HuZx · 2025-11-26
>
> The evaluation of this submission has a large variance so we need to gather more information from both the authors and the reviewers. Please take a look at the author's response and see how it affects your evaluation.
>
> Best,
> AC

---

> > ### Author Response · Authors · 2025-11-27
> >
> > Dear reviewer taWz,
> >
> > Thank you for your review and valuable feedback. We have provided detailed responses and additional experiments. As the discussion phase is ending, we would greatly appreciate it if you could review our rebuttal. We are willing to address any further concerns during the remaining rebuttal period. Thank you again for your time and consideration.
> >
> > Sincerely,
> >
> > Authors

---

### Official Review · Reviewer_CfZM · 2025-11-05

**Soundness:** 3
**Presentation:** 3
**Contribution:** 4
**Rating:** 8
**Confidence:** 3

**Summary:**

The paper introduces Regret-Guided Search Control (RGSC), an extension to AlphaZero that enhances learning efficiency in reinforcement learning for board games by prioritizing restarts from high-regret states—positions where the agent's evaluation significantly diverges from the actual game outcome. RGSC incorporates a regret network (comprising a ranking network and a value network) to identify these states from self-play trajectories and MCTS nodes, storing them in a prioritized regret buffer for reuse as starting points.

Contributions include a novel regret-based ranking objective to handle imbalanced and non-stationary regret distributions, a toy example demonstrating regret-guided benefits, and empirical results showing RGSC outperforms AlphaZero and Go-Exploit by 77 and 89 Elo on average across 9x9 Go, 10x10 Othello, and 11x11 Hex. Additionally, RGSC improves performance on nearly converged models (e.g., boosting win rate against KataGo from 69.3% to 78.2%), highlighting its robustness.

**Strengths:**

The paper demonstrates strong originality by creatively combining search control concepts (e.g., from Sutton & Barto, 2018, and Go-Exploit) with a novel regret-guided prioritization mechanism tailored to AlphaZero. This includes a ranking-based objective for the regret network, which addresses challenges like imbalance and non-stationarity in regret prediction—issues not fully tackled in prior work like Tavakoli et al. (2020) or Trudeau & Bowling (2023). While building on existing ideas, the application to identify high-impact states in MCTS trees and self-play trajectories removes limitations of uniform sampling, offering a fresh formulation inspired by human learning patterns.

In terms of quality, the work is rigorous, with comprehensive experiments on three diverse board games using fixed training states for fair comparison, substantiated by Elo ratings, error bars, and additional analysis. The methodology includes derivations and pseudocode in appendices, enhancing reproducibility.

Clarity is excellent; the paper is well-structured, with intuitive figures.

Significance is high, as RGSC could extend to broader RL domains (e.g., robotics or planning) where sample efficiency is critical.

**Weaknesses:**

1. Adding the regret network increases model complexity (e.g., extra heads and training objectives), but the paper doesn't quantify the overhead in terms of GPU hours or inference time during self-play. This could be addressed by reporting relative costs compared to baselines, ensuring the efficiency gains aren't offset by higher per-iteration expenses.

2. While RGSC shows final Elo improvements, the Elo curves in Figure 4 reveal that advantages are not consistently stable across training, with baselines often close and fluctuations (e.g., dips around 50K-60K steps) raising concerns about robustness.

**Questions:**

1. Given the fluctuations and closeness in Figure 4's Elo curves (e.g., overlaps in confidence intervals), how stable is RGSC's advantage? Could you report results from extended training (e.g., to 70K-100K steps) or more random seeds to assess if the lead holds or diminishes over time?
2. Same question for Figure 6.
3. Could you report the relative computational cost of RGSC (e.g., average time per self-play game or optimization step) compared to AlphaZero and Go-Exploit? How much overhead does the regret network introduce, and does it offset the sample efficiency gains?
4. The toy example uses a simple regret ($|\hat Q(s) - Q(s,a)|$); how does this compare to the definition in Equation 2?
5. I find the RGSC method to be very strong. Could the authors discuss the potential for applying RGSC in more complex domains, specifically those involving stochastic dynamics or partial observability? What challenges would the regret network face in these settings?

---

> ### Author Response · Authors · 2025-11-22
> **Response to Reviewer CfZM**
>
> Thank you for your positive comments and for providing insightful feedback. We address each weakness and question below.
>
> > W1 & Q3: the relative computational cost of RGSC
>
> Please refer to our general response on "Regarding the computational cost of RGSC".
>
> > W2 & Q1 & Q2: report results from extended training
>
> Regarding Figure 4, each game already reports the average over two independent training runs to reduce noise in the main text. To further verify the stability of RGSC, we extended all three experiments in Figure 4 from 60k to 80k steps (due to limited computational resources, we were only able to add an additional 20k steps). As in the original setting, we evaluate every 3k steps. The extended results below show that RGSC consistently achieves higher Elo ratings than both AlphaZero and Go-Exploit across all three games, demonstrating the robustness of our method.
>
> |9x9 Go|60k|63k|66k|69k|72k|75k|78k|
> |-|-|-|-|-|-|-|-|
> |AlphaZero|1214.3|1213.1|1228.7|1216.3|1215.7|1229.8|1229.7|
> |Go-Exploit|1194.2|1206.9|1191.5|**1221.0**|1223.5|1236.5|1226.7|
> |RGSC|**1289.5**|**1247.8**|**1299.9**|1212.7|**1281.7**|**1366.3**|**1307.1**|
>
> |10x10 Othello|60k|63k|66k|69k|72k|75k|78k|
> |-|-|-|-|-|-|-|-|
> |AlphaZero|1283.8|1280.3|1299.0|1327.9|1311.4|1320.3|1332.3|
> |Go-Exploit|1310.2|**1333.8**|1334.6|1352.1|**1368.5**|1361.5|1357.0|
> |RGSC|**1360.3**|1319.2|**1349.6**|**1358.5**|1365.4|**1390.7**|**1369.9**|
>
> |11x11 Hex|60k|63k|66k|69k|72k|75k|78k|
> |-|-|-|-|-|-|-|-|
> |AlphaZero|1421.2|1363.2|1393.3|1386.8|1380.0|1409.9|1433.2|
> |Go-Exploit|1383.2|1410.2|1394.7|1428.9|1437.4|1431.7|1463.2|
> |RGSC|**1504.8**|**1456.1**|**1485.8**|**1500.4**|**1485.8**|**1442.8**|**1518.5**|
>
> Regarding Figure 5, we also extended the training steps from 40k to 70k. As shown below, RGSC maintains a stable lead over both baselines throughout the extended training period, further demonstrating its robustness. Note that the evaluation is conducted every 5k steps, using the same settings as in the original Figure 5.
>
> |Go|40k|45k|50k|55k|60k|65k|70k|
> |-|-|-|-|-|-|-|-|
> |AlphaZero|1238.0|1164.1|1219.4|1202.2|1219.0|1215.9|1163.3|
> |Go-Exploit|1162.9|1194.3|1195.5|1252.2|1182.8|1225.3|1215.3|
> |RGSC|**1250.0**|**1208.6**|**1306.3**|**1280.9**|**1293.1**|**1271.8**|**1242.8**|
>
> Regarding Figure 6, due to the high computational demand, we are unable to extend the experiment currently. We are still running these experiments and will update the results as soon as they become available.
>
> Overall, the extended results show that RGSC maintains a stable performance advantage over the baselines. We hope these additional experiments address your concerns.
>
> > Q4: The toy example uses a simple regret $\lvert \hat{Q}(s)-Q(s, a)\rvert$; how does this compare to the definition in Equation 2?
>
> We use a simple regret definition in the toy example to illustrate the core idea of regret-guided search control in a minimal setting. In AlphaZero, we found that the regret in Equation (2) provides a more suitable regret signal because the board game outcomes are binary (0 or 1) without any intermediate rewards. Although the formulas differ, both capture the same principle: providing a signal that prioritizes states where the agent's evaluation deviates most from the target outcome.
>
> > Q5: the potential for applying RGSC in more complex domains
>
> Please refer to our general response on "Regarding applying RGSC to other environments beyond board games".
>
> Thank you again for your positive comments. We hope our additional experiments and clarifications address your concerns.

---

> ### Author Response · Authors · 2025-11-25
> **Extending Training for Figure 6**
>
> Regarding Figure 6, we have finished the extension from 60k to 80k steps and evaluated every 3k steps. Note that both regret ranking head and regret value head are introduced in this paper. Overall, the results show that the selected nodes guided by the regret ranking head still consistently achieve higher Elo ratings than those guided by the regret value head across all three games. These findings suggest that the ranking head is more effective in identifying candidates with greater learning potential.
>
> |9x9 Go|60k|63k|66k|69k|72k|75k|78k|
> |-|-|-|-|-|-|-|-|
> |Regret Value Network|1185.45|1181.17|1202.84|**1229.33**|1222.51|1207.01|1195.84|
> |Regret Ranking Network|**1289.5**|**1247.8**|**1299.9**|1212.7|**1281.7**|**1366.3**|**1307.1**|
>
> |10x10 Othello|60k|63k|66k|69k|72k|75k|78k|
> |-|-|-|-|-|-|-|-|
> |Regret Value Network|1310.39|1253.56|1315.22|1349.27|1328.06|1333.07|1353.58|
> |Regret Ranking Network|**1360.3**|**1319.2**|**1349.6**|**1358.5**|**1365.4**|**1390.7**|**1369.9**|
>
> |11x11 Hex|60k|63k|66k|69k|72k|75k|78k|
> |-|-|-|-|-|-|-|-|
> |Regret Value Network|1449.35|1391.57|1449.35|1432.30|1412.80|**1472.92**|1405.48|
> |Regret Ranking Network|**1504.8**|**1456.1**|**1485.8**|**1500.4**|**1485.8**|1442.8|**1518.5**|
>
> We hope these experiments address your concerns. If there is anything further we can do, please let us know. We are willing to engage in any further discussions or conduct additional experiments as needed during the rebuttal period. Thank you again for your time and effort.

---

> > ### Comment · Reviewer_CfZM · 2025-11-25
> >
> > Thank you for the extra experiments, these have resolved my concerns. I acknowledge your contributions and will maintain the original rating.

---

> > > ### Author Response · Authors · 2025-11-27
> > >
> > > Thank you very much for your kind response and for increasing the confidence score. We truly appreciate your support! If there is anything further we can do, please let us know. Thank you again for your time and effort.

---

### Author Response · Authors · 2025-11-22
**General Response to All Reviewers**

Dear all reviewers,

We sincerely thank all reviewers for their time and effort in providing insightful and constructive feedback. Below, we address the major concerns raised by the reviewers.

- ### Regarding **applying RGSC to other environments beyond board games**. (Reviewers CfZM, taWz, and 4XJP)

Our motivation is to apply search control within the AlphaZero algorithm, whose standard benchmarks are board games. Therefore, our experiments follow this established evaluation setting. However, RGSC is not restricted to board games. The method does not use any game-specific assumptions; it simply adds two heads and leverages states from MCTS and self-play. Thus, RGSC naturally applies to any AlphaZero-style algorithm. For example, many successors of AlphaZero use the same architecture–policy/value networks+MCTS–across diverse environments, such as MuZero (Atari), Stochastic MuZero (stochastic domains), and Sampled MuZero (continuous control). We believe RGSC can be integrated into these algorithms with only minor modifications. **To demonstrate this, we applied RGSC to MuZero on Pac-Man, one of the Atari games. Under the same training budget, RGSC-MuZero reaches 5166 points, compared to 3704 for MuZero.** This confirms that RGSC improves learning efficiency beyond board games, showing its potential for broader applicability. This experiment is included in Appendix H of the revision, and we also summarize the above sentences in the discussion section.

- ### Regarding the **choice of hyperparameters** in RGSC. (Reviewers taWz and WKT8)

We would like to clarify that RGSC introduces only four additional hyperparameters, including the sampling probability $\lambda$, the buffer sampling temperature $\tau$, the buffer size $\kappa$, and the EMA coefficient $\alpha$. All other parameters shown in Table 2 are inherited directly from the original AlphaZero algorithm and were directly taken from prior works.

To examine whether RGSC is sensitive to these hyperparameters, we added additional experiments in Appendix E of the revision, where we compare the performance under different hyperparameter settings. Specifically, we tested $\lambda \in \\{0.2, 0.5, 0.9\\}$, $\tau \in \\{0.1, 0.5, 1.0\\}$, $\kappa \in \\{100, 500, 1000\\}$, and $\alpha \in \\{0.1, 0.5, 1.0\\}$. From the results (please refer to Appendix E in the revision), the hyperparameters used in the main paper ($\lambda=0.5, \tau=0.1, \kappa=100, \alpha=0.5$) achieve the overall best performance across all games. **These experiments show that RGSC is not highly sensitive to hyperparameter choices and that our recommended configuration works well across different games, demonstrating the overall robustness of the method.**

- ### Regarding the **computational cost** of RGSC (Reviewers CfZM and 4XJP)

Although RGSC adds two additional heads (regret value and ranking heads), they share the same backbone as the policy/value network, so the additional computation is minimal, especially in larger models. To quantify this, we measured both the neural network inference time and the per-iteration wall-clock time on Go for the 3-block model (used in Section 4.2) and the 15-block model (used in Section 4.3). For the 3-block model, RGSC is 1.35x slower in inference and 1.25x slower per iteration compared to AlphaZero/Go-Exploit. However, for the 15-block model, RGSC is approximately 1.03x slower in both inference and per-iteration compared to AlphaZero/Go-Exploit, which is nearly identical. In realistic settings (e.g., AlphaZero used 20 blocks and KataGo used 20-40 blocks), this overhead becomes negligible. Therefore, **RGSC adds almost no extra cost while improving training efficiency in practice.** We appreciate the reviewers' suggestion and have included the above description in Appendix A.1.

---

In addition, we have uploaded a revised version based on the reviews. To make it easier to identify the changes, all revisions are highlighted in blue text. Below is a summary of the updates:

* [Figure 7] Included uncertainty bands from two independent runs. (Reviewer WKT8)
* [Discussion] Added sentences on broader applications of RGSC. (Reviewers CfZM, taWz, and 4XJP)
* [Appendix A.1] Added a paragraph discussing the computational cost of RGSC. (Reviewers CfZM and 4XJP)
* [Appendix E] Added experiments for evaluating the performance under different RGSC hyperparameter settings. (Reviewers taWz, and WKT8)
* [Appendix H] Added experiments applying RGSC to MuZero on Atari games. (Reviewers CfZM, taWz, and 4XJP)
* We have also revised all other typos and wording suggestions provided by reviewers accordingly.

We hope the additional experiments and revisions address your concerns and improve the clarity and quality of the paper. These results show that RGSC is robust and broadly applicable, representing a meaningful and valuable step forward in improving search-control efficiency for AlphaZero-style methods.

---

### Meta-Review · Area_Chair_MatW · 2026-01-06

**Summary:**

The dispersion of scores for this paper was unusual, with two reviews (very) positive and two reviews very negative. To further complicate matters, the two negative reviewers did not participate in the discussion, despite repeated attempts to engage them by the former area chair and the authors themselves.

Overall, reading the negative reviews and the rebuttal, I have the impression that the negative reviews do not raise fundamental issues. The rebuttal of the authors did a good job overall of defending the paper.

While there is always a possibility for a paper to improve, experiment on more benchmarks, and show even higher gains over baselines, I think this paper clears the bar for being a conference contribution. Given the lack of engagement from the negative reviewers, I'm leaning on the side of the authors.

**Reviewer Concerns:**

Reviewer WKT8 claims that the plots in the paper do not show a clear advantage of the proposed method over the existing baselines. The author's response on this point seemed quite reasonable to me. Furthermore, I find that overemphasizing benchmark numbers is a dangerous way to approach contributions. Despite attempts from the previous area chair and from the authors, the reviewer did not engage in a discussion.

Reviewer taWz found the choice of benchmarks weak and had reservations about the experimental setup (especially the choice of hyperparameters). Again, I thought the authors' answer was reasonable and on point. In general, there didn't seem to be any serious questions about correctness or reproducibility. Despite attempts by the previous area chair and the authors, the reviewer did not engage in discussion.

**Reviewer Scores:**

It is hard to predict how the negative reviewers, who did not engage in discussion, would have reacted to the authors' rebuttal.

The positive reviewers, on the other hand, were appreciative of the authors' additional experiments and explicitly mentioned maintaining a positive opinion of the contribution.

---

### Decision · Program_Chairs · 2026-01-26

Accept (Poster)